# Maintenance of sarcomeric integrity in adult muscle cells crucially depends on Z-disc anchored titin

Sandra Swist [1✉], Andreas Unger[2], Yong Li[2], Anja Vöge[1], Marion von Frieling-Salewsky[2], Åsa Skärlén[3], Nicola Cacciani[4], Thomas Braun [5], Lars Larsson [4] & Wolfgang A. Linke [2✉]

The giant protein titin is thought to be required for sarcomeric integrity in mature myocytes, but direct evidence for this hypothesis is limited. Here, we describe a mouse model in which Z-disc-anchored TTN is depleted in adult skeletal muscles. Inactivation of TTN causes sarcomere disassembly and Z-disc deformations, force impairment, myocyte de-stiffening, upregulation of TTN-binding mechanosensitive proteins and activation of protein quality-control pathways, concomitant with preferential loss of thick-filament proteins. Interestingly, expression of the myosin-bound Cronos-isoform of TTN, generated from an alternative promoter not affected by the targeting strategy, does not prevent deterioration of sarcomere formation and maintenance. Finally, we demonstrate that loss of Z-disc-anchored TTN recapitulates muscle remodeling in critical illness 'myosinopathy' patients, characterized by TTN-depletion and loss of thick filaments. We conclude that full-length TTN is required to integrate Z-disc and A-band proteins into the mature sarcomere, a function that is lost when TTN expression is pathologically lowered.

[1] Department of Systems Physiology, Ruhr University Bochum, D-44780 Bochum, Germany. [2] Institute of Physiology II, University of Munster, D-48149 Munster, Germany. [3] Department of Clinical Neuroscience, Clinical Neurophysiology, Karolinska Institute, SE-171 77 Stockholm, Sweden. [4] Department of Physiology and Pharmacology, Karolinska Institute, SE-171 77 Stockholm, Sweden. [5] Department of Cardiac Development and Remodeling, Max Planck Institute for Heart and Lung Research, D-61231 Bad Nauheim, Germany. ✉email: sandra.swist@ruhr-uni-bochum.de; wlinke@uni-muenster.de

Skeletal muscles are required for gait, posture, and many additional functions. As much as 80–90% of the volume of an adult skeletal muscle cell is occupied by the contractile units known as the sarcomeres. These structures are multi-protein assemblies of paracrystalline order, which essentially consist of three filament systems bordered by Z-discs: actin-based thin filaments, myosin-based thick filaments, and titin filaments. The giant titin proteins (TTN) span half-sarcomeres from the Z-disc to the M-band[1] and overlap with TTN proteins from adjacent half-sarcomeres[2]. Several different alternative transcripts are generated by the *TTN* gene, including full-length isoforms (3–3.8 MDa) termed N2B, N2BA (in the heart) and N2A (in skeletal muscles), as well as the small, Z-disc-anchored Novex-3 isoform (~0.7 MDa)[3,4]. Moreover, the recently discovered Cronos isoform arises from an alternative transcription start site >150 kilobases (kb) downstream of the canonical promoter, in *Ttn* intron 239[5]. Ever since its discovery, TTN has been thought to hold the sarcomere together and provide it with stability and elasticity[6,7]. The role of TTN for the elasticity and stretch-dependent passive tension of myocytes is now well established[8,9]. Titin-based elastic force also complements and regulates the contractile force generated by actin and myosin[10,11]. Furthermore, TTN determines the length of the sarcomeric thick filaments[12] and is important for de-novo sarcomere assembly[13–17]. It has also been frequently suggested that TTN is relevant for the maintenance of the mature sarcomere. This function has been more difficult to assess and details have remained unclear. However, TTN is necessary for cardiac development and its absence leads to early embryonic lethality[13,14,18].

Here, we report an inducible, conditional TTN-knockout (KO) mouse model targeting the TTN isoforms expressed from the canonical *Ttn* promoter in adult skeletal muscles. Our approach differs from previous strategies, because earlier TTN-deficient rodent models either preserved the structural continuity of the protein, e.g., by deleting only the cardiac N2B-element[19], certain I-band immunoglobulin (Ig) domains[20,21], the PEVK-segment[22,23], or the distal M-band titin[15,16,24], or they were designed to partially or completely eliminate TTN at early stages of development[14–16,18,25]. In our model, we find that inactivation of N2A and Novex-3 TTN in adult skeletal muscles reduces muscle mass, contractile strength, and myocyte stiffness, causes sarcomeric disintegration along with Z-disc aggregation and streaming, and results in preferential loss of thick-filament proteins. TTN-depletion also causes upregulation of titin-binding proteins involved in mechanotransduction and activates components of the protein quality control (PQC) machinery, providing additional insight into the role of TTN in sarcomere maintenance and protein turnover. Strikingly, expression of Cronos is unable to prevent cardiac embryonic lethality or disassembly of adult skeletal muscle sarcomeres. Our findings prove the long-hypothesized crucial function of TTN for protein homeostasis in mature muscle sarcomeres.

Interestingly, our conditional TTN KO-mouse model recapitulates key changes observed in the skeletal muscles of critically ill intensive care unit (ICU) patients with acute quadriplegic myopathy (critical illness myopathy (CIM) or myosinopathy), which is seen in up to ~30% of ICU patients[26]. A hallmark of CIM is the preferential loss of myosin and myosin-associated proteins but not thin-filament proteins in the sarcomeres of proximal and distal skeletal muscles[27]. We find that CIM patient muscles show not only preferential loss of thick filaments but also suffer from reduced TTN content, while sarcomeric actin and the major thin-filament protein nebulin are largely preserved. Our results suggest that downregulation of TTN is a contributing factor in the pathogenesis of CIM, and presumably also in other types of muscle atrophy.

## Results

**Full-length titin is crucial for de-novo sarcomerogenesis**. We initially generated titin 'knockout-first' heterozygous Ttn[tm1a]/+ mice with conditional potential, targeting the N-terminus of TTN (Supplementary Fig. 1, Supplementary Fig. 2, Supplementary Fig. 3). Heterozygous mice were healthy but no homozygous mutant mice were born, because titin-deficient embryos died around E10 (Supplementary Fig. 2). Interestingly, expression of Cronos, which was not targeted by our approach, was unaltered in mutant embryos (Supplementary Fig. 2). Therefore, Z-disc-integrated TTN is required for cardiac sarcomerogenesis and Cronos does not rescue cardiac embryonic lethality.

**Inducible skeletal muscle-specific *Ttn* removal in adult mice**. Next, we generated the conditional Ttn[tm1c] allele by deleting the lacZ reporter and neomycin selection cassettes using FLP recombinase, preventing embryonic lethality of the Ttn[tm1a] mice (Fig. 1a, b). Homozygous Ttn[tm1c/tm1c] animals were fertile and had no obvious phenotype. Breeding of Ttn[tm1c/tm1c] animals with ACTA1-rtTA;tetO-cre mice[28], in which CRE-expression is driven by reverse tetracycline-controlled transactivator (rtTA) under the human alpha skeletal muscle actin 1 (ACTA1) promoter, enabled us to delete *Ttn* exons 4–6 in adult skeletal myocytes upon doxycycline (dox) administration (Fig. 1a). To monitor the effectiveness of dox-treatment, we studied Rosa26-LacZ/Tg (ACTA1-rtTA,tetO-cre) reporter mice, which showed strong ß-galactosidase staining in fatigue-resistant (diaphragm, soleus) and fast skeletal muscles, but not in heart, lung or liver after 30 days of treatment (Supplementary Fig. 4).

Inactivation of *Ttn* after 30 days of dox-treatment was verified by RT-PCR, using primers to exons 2 and 8 (Fig. 1c). Deletion of *Ttn* exons 4-6 induced a frameshift leading to premature stop codons, which resulted in loss of full-length *Ttn* mRNA by nonsense-mediated decay and loss of TTN protein (Fig. 1d). To rule out the generation of new TTN isoforms with alternative N-termini from the mutant allele[18], we performed quantitative RT-PCR. Expression of *Ttn* exons 3–4 and 20–21 was reduced by 99% and ~84%, respectively, in MUT vs. WT gastrocnemius muscles (Fig. 1e). In contrast, expression of Cronos remained unaltered in MUT mice, whereas *Ttn* exons 239–240 flanking the Cronos promoter region showed the anticipated ~90% reduction (Fig. 1e). Likewise, in situ hybridizations with a riboprobe detecting *Ttn* exon 20–24 transcripts clearly demonstrated loss of WT *Ttn* mRNA (Fig. 1f).

**TTN loss causes muscle atrophy and reduces muscle strength**. Profound weight loss began in Ttn[tm1c/ltm1c];ACTA1-rtTA[pos];tetO-cre[pos] MUT mice ~3 weeks after initiating the dox-treatment (Fig. 2a). MUT mice had 7.1 ± 1.8% (mean ± SD) lower body weight than WT mice ($p = 0.0026$; WT, $n = 10$; MUT, $n = 11$) on day 21 of dox-treatment, and 17 ± 1.9% ($p < 0.001$; WT, $n = 9$; MUT, $n = 10$) lower weight on day 30 (Fig. 2a, Supplementary Fig. 5a). Individual muscles already lost weight in some mice, even when whole body weights were still unaltered (Supplementary Fig. 5b). The experiment was terminated, and tissues were collected, when animals had lost maximally 20% of their initial weight.

Dox-treated MUT mice performed significantly worse than control animals in a four-limb hanging test (4LHT) (Fig. 2b, Supplementary Fig. 5c). MUT mice showed reduced hang time after 24 days of dox-treatment (6.8 ± 1.5 min (mean ± SD), versus ~12.5 min before treatment ($p = 0.0004$, $n = 7$)). On day 31 of treatment, the hang time was reduced to only ~20% of the initial value (Fig. 2b).

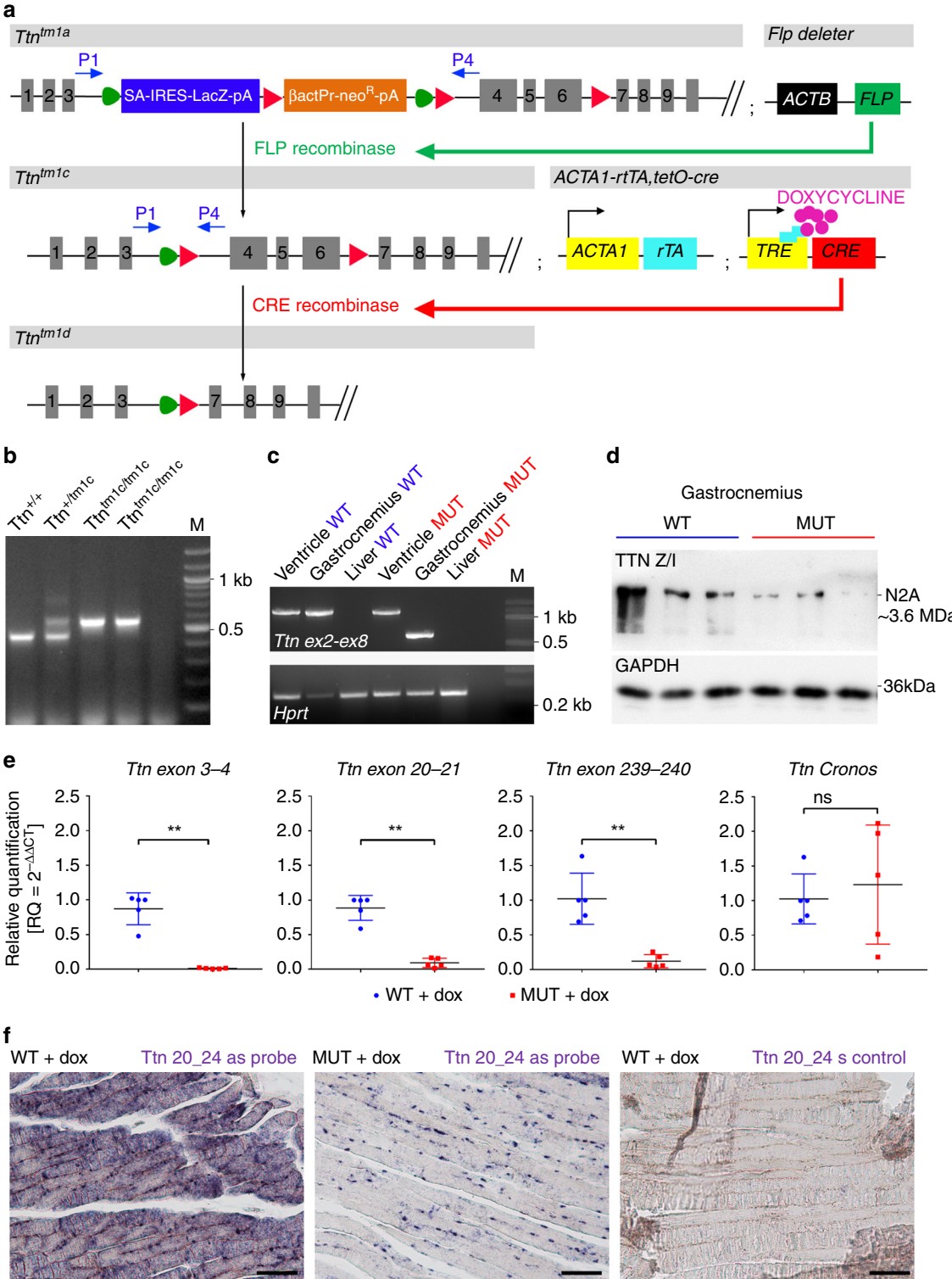

Figure showing panels a–f as described.

Histological analysis by hematoxylin-and-eosin staining of skeletal muscle revealed reduced myofiber diameters and some centralized nuclei in MUT (Fig. 2c), indicating muscle atrophy. Both the number of centralized nuclei (Fig. 2d) and the mean fiber cross-sectional area (Fig. 2e) were already altered in MUT muscles after 21 days of dox-treatment.

**TTN depletion disrupts the normal striated sarcomere pattern.** Immunostaining against different TTN epitopes (Fig. 3a) on paraffin sections of M. gastrocnemius showed increased

disorder and reduced signal intensities in MUT vs. WT myocytes (Fig. 3b–f). Antibodies to the TTN Z/I-junction (Z/I-2080) demonstrated protein loss or reduction in many myocytes within the same muscle (Fig. 3b). The proportion of cells lacking regular Z-disc protein striations reached ~20% and ~40% after 21 days and ~30 days, respectively, of dox-treatment (Fig. 3c). Immunostaining with Novex-3-specific antibodies confirmed the absence of this TTN isoform in most MUT cells (Fig. 3d). Loss of TTN was always accompanied by disruption of the striated patterns of Z-disc-based α-actinin

**Fig. 1 Inducible skeletal muscle-specific titin knockout. a** Mutant Ttn$^{tm1a}$ allele generation, with the reporter/selection cassette inserted upstream of *Ttn*-exons 4–6. This cassette contains an FRT-site (green triangles), followed by a LacZ-trapping element (blue square) with a strong splice acceptor (SA) and internal ribosomal entry site (IRES). LoxP-sites are indicated by red triangles. The first loxP-site is followed by neomycin (neo$^R$) under the control of the human β-actin promoter (βactPr), a simian virus terminator and polyadenylation signal (pA), a second FRT-site, and a second loxP-site. A third loxP-site is inserted downstream of the critical exons. The selection and lacZ cassettes were deleted by FLP recombinase-expressing mouse strain (Flp-deleter). Dox-induced CRE-recombinase expression in the hybrid line Ttn$^{tm1c/ltm1c}$;ACTA1-rtTA;tetO-cre mediated recombination of the remaining loxP-sites and removed exons 4–6 (frameshift). **b** Agarose gels resolving genotyping PCR products from WT (Ttn$^{+/+}$), heterozygous (Ttn$^{+/tm1c}$) and homozygous MUT (Ttn$^{tm1c/tm1c}$) mice. PCR amplicons: WT, 450 bp; mutant Ttn$^{tm1c}$-allele, 550 bp. **c** Agarose gels resolving products of RT-PCR with RNA from different tissues of dox-treated WT and MUT Ttn$^{tm1c}$;ACTA1-rtTA;tetO-cre mice. Primers spanning exons 2–8 (1.25 kb) revealed a 0.6 kb amplicon in gastrocnemius but not cardiac (ventricle) muscle, demonstrating skeletal muscle-specific loss of exons 4–6. **d** TTN N2A-protein expression in WT and MUT gastrocnemius detected by western blot using antibodies to Z/I-band titin (2080). Loading control was GAPDH. Gels shown in b) are representative of >5 gels/condition, in **c** and **d**) 2/condition. **e** Quantitative RT-PCR analysis proving loss of full-length TTN RNA in dox-treated MUT vs. WT mice, using primers to exons 3–4 ($p = 0.0079$) and exons located 3′ to the deletion, specifically, 20–21 ($p = 0.0079$) and 239–240 ($p = 0.0079$); the alternative Cronos *Ttn* transcript remained unaltered ($p = 0.7937$). $n = 5$ animals/group; data are mean ± SD. \*\*$p < 0.01$; ns non-significant in unpaired, two-tailed Mann–Whitney *t*-test. **f** RNA in situ hybridizations (representative of 2–4 images/condition) with antisense (as) riboprobe targeting *Ttn*-exons 20–24, which detected *Ttn*-RNA in both cytoplasm and nuclei of WT but only in nuclei of MUT cells. Sense (s) *Ttn*-exons 20–24 riboprobe served as negative control. Bars, 100 μm. All data were obtained after ~30 days of dox-treatment. Source data are provided as a Source Data file.

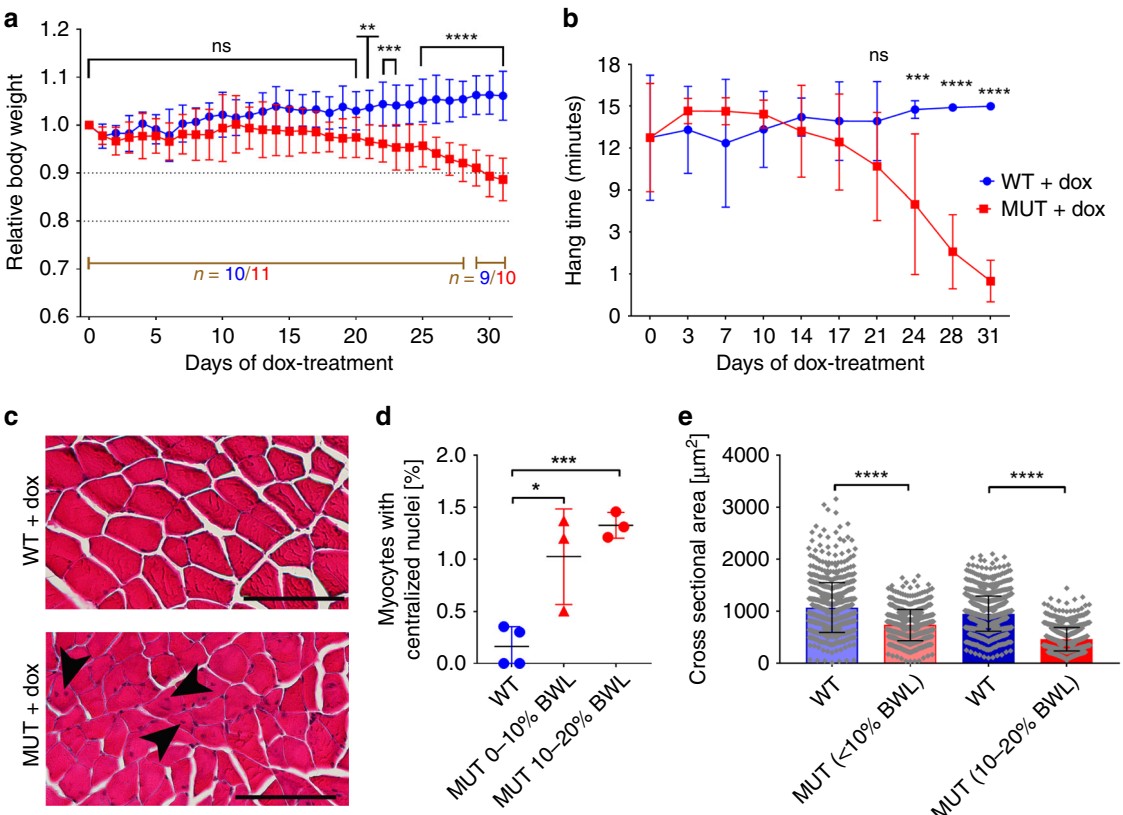

**Fig. 2 Titin depletion leads to muscle remodeling and reduction in muscle strength. a** Alterations in body weight with time in MUT and WT Ttn$^{tm1c}$;ACTA1-rtTA;tetO-cre mice during dox-treatment. Data were normalized to the body weight immediately before dox-treatment. Beginning around day 21, there was a significant difference between WT and MUT ($p = 0.0068$; for WT, $n = 10$ animals (9 for the last three time points) and for MUT, $n = 11$ animals (10 for the last three time points)). **b** Alterations in muscle strength during dox-treatment of MUT *vs.* WT mice measured in the 4-limb hanging test. Hang time was plotted against duration of treatment. From day 24 of dox-treatment onward, MUT mice had significantly reduced hang times ($p = 0.0004$; $n = 7$ animals/group). **c** Representative cross sections (out of 8–10 images recorded per condition) of hematoxylin and eosin stained M. triceps brachii from WT and MUT mice after ~30 days of dox-treatment. Arrowheads point to centralized nuclei. Bars, 100 μm. **d** Percentage of myofibers presenting with centralized nuclei in dox-treated WT ($n = 4$) and MUT mice with a body weight loss (BWL) of <10% ($n = 3$) and 10–20% ($n = 3$), respectively, relative to body weight before treatment. **e** Cross-sectional area of myofibers ($n = 630$) from WT and MUT Ttn$^{tm1c}$;ACTA1-rtTA;tetO-cre mice (~30 days of dox-treatment); $n = 3$ mice/group. Data are mean ± SD. \*\*\*\*$p < 0.0001$; \*\*\*$p < 0.001$; \*\*$p < 0.01$, in 2-way-ANOVA, followed by Sidak's multiple comparisons test (**a** and **b**); unpaired, two-tailed Student's *t*-test (**d**); or two-tailed Mann–Whitney test (**e**). Source data are provided as a Source Data file.

(ACTN2, Fig. 3b, e, f) and A-band-based myosin heavy chain (MyHC) (Fig. 3d). Furthermore, both the TTN-kinase domain (M-band) and the Cronos N-terminus (I/A-junction) lost their regular striated appearance in affected MUT cells (Fig. 3e,

f). Thus, the highly ordered repetitive sarcomere structure depends on the presence of Z-disc-anchored TTN isoforms, whereas Cronos does not prevent deterioration of sarcomeres.

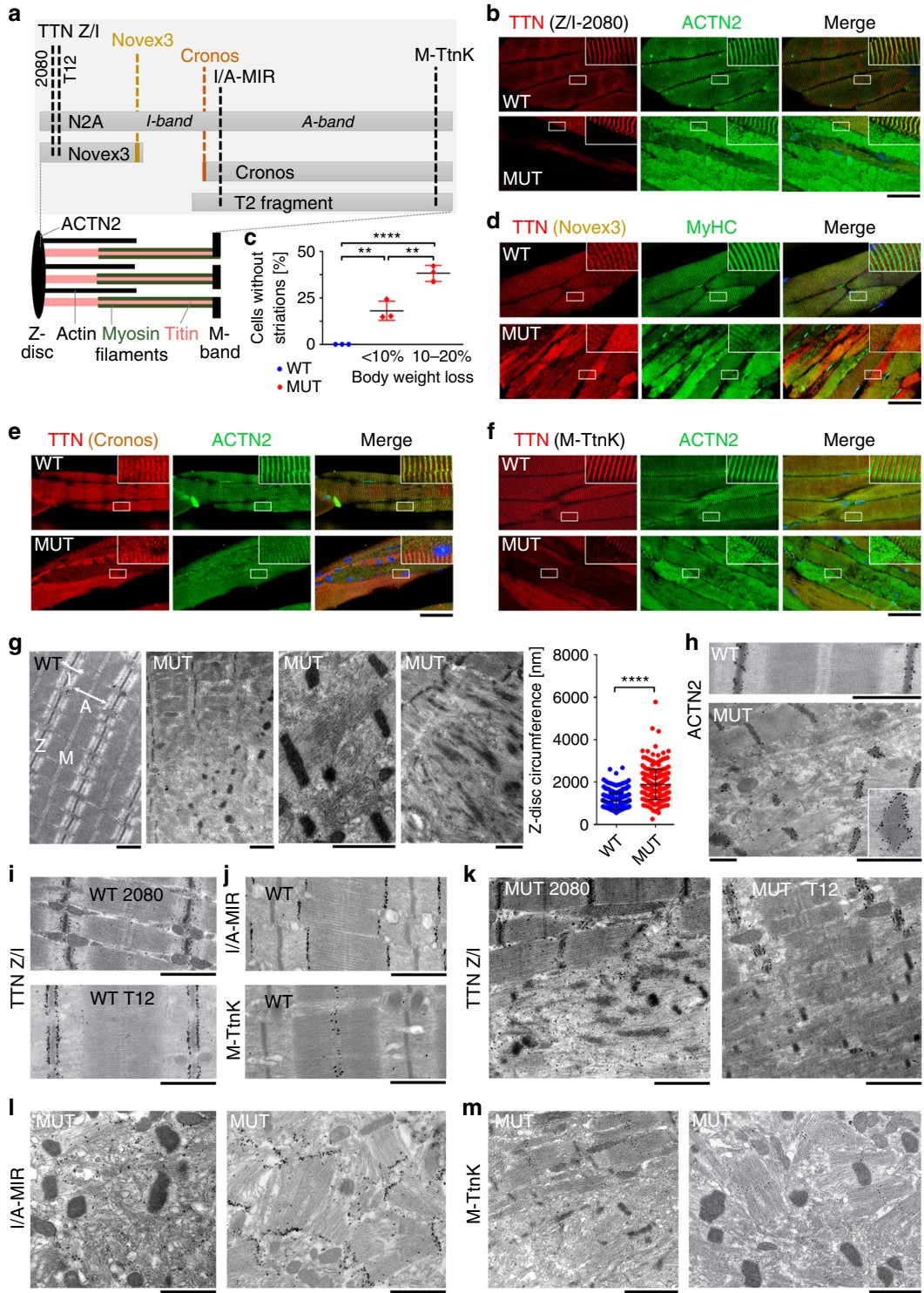

**TTN loss disconnects A- and Z-bands and promotes aggregation.** Electron micrographs of fast-type M. gastrocnemius revealed severe ultrastructural changes in TTN-depleted myocytes after ~30 days of dox-treatment, while yet-unaffected MUT cells still showed the regular, WT-like sarcomere pattern (Fig. 3g). In affected myocytes, we observed the following alterations: (1) Sarcomere disintegration, typically beginning at the I-bands, including dissociation of sarcomeric Z-discs and A-bands, which became misaligned relative to one another. The organization into I-, A-, and M-bands disappeared. (2) Loss of the parallel orientation of myofibrils. (3) Appearance of Z-discs as thickened rod-shaped or irregularly shaped structures with increased average circumference (Fig. 3g); Z-disc shrinking or extensive streaming were frequent. (4) Despite the continuous loss of thick filaments, A-bands were sometimes well-preserved. Moreover, we observed cells with prominent morphological aberrations in one region, while other regions still contained normal sarcomeres, and profoundly altered myocytes next to cells that still appeared healthy (Fig. 3g). Similar patterns of disruption were already present in MUT muscles after 21 days of

**Fig. 3 Ultrastructural changes in dox-treated MUT vs. WT gastrocnemius myocytes. a** Anti-TTN antibody epitope locations and TTN variants. **b** Confocal images of immunolabeled longitudinal sections of WT and MUT muscles after 30 days of dox-treatment, using antibodies to titin Z/I-band (2080; red) and α-actinin (ACTN2, green); nuclear staining (Hoechst, blue) shown on merge. **c** Percentage of cells without cross-striations in WT and MUT gastrocnemius muscles after 21 days (<10% body weight loss, BWL) or >28 days of dox-treatment (>10% BWL), counted on images stained against Ttn-Z/I-2080 and ACTN2; $n = 3$ animals/group. **d**–**f** Confocal images of immunolabeled longitudinal sections of WT and MUT muscles after 30 days of dox-treatment, using antibodies to various TTN species (red) and ACTN2 (green) or myosin heavy chain (MyHC, green); nuclear staining by Hoechst (blue) is also shown on merged images. **d**) TTN Novex3 and MyHC co-staining. **e**) TTN Cronos and ACTN2 co-staining. **f**) TTN kinase domain near M-band (M-TtnK) and ACTN2 co-staining. In **b** and **d**–**f**, secondary antibodies are Cy3- or fluorescein-conjugated anti-rabbit or anti-mouse IgG. Bars, 25 μm; insets show detail at 3x magnification. **g** Electron micrographs of WT (left panel) and MUT muscles after 30 days of dox-treatment. Z, Z-disc; I, I-band; A, A-band; M, M-band. Right panel shows average circumference of sagittal Z-disc planes, measured on electron micrographs of doxycycline-treated WT and MUT mouse (>10% BWL) muscles ($n = 2$ animals/group and 250 Z-discs/group). **h** Immunoelectron micrographs using antibodies to ACTN2. Inset shows detail at ~2.5x magnification. **i**–**m** Immunoelectron micrographs using antibodies to four different TTN epitopes. **i** Antibody epitopes in WT. **k** Localization of TTN Z/I-band junction (TTN Z/I) by 2080 and T12 antibodies in MUT. **l** Localization of TTN I/A-band junction by MIR antibody (I/A-MIR). **m** Localization of TTN M-band region by TtnK antibody (M-Ttnk). In **g**–**m**, secondary antibodies are nanogold-conjugated anti-mouse or anti-rabbit IgGs; bars, 1 μm. Data in **c** and **g** are mean ± SD; ****$p < 0.0001$; **$p < 0.01$, in unpaired, two-tailed Student's $t$-test (**c**) or two-tailed Mann–Whitney test (**g**). Immunofluorescence micrographs shown are representative of 2–5 images/condition, electron micrographs of >25 images/condition. Source data are provided as a Source Data file.

dox-treatment (Supplementary Fig. 6a), although at a much lower degree.

Immunogold-EM demonstrated that Z-disc-like rods/aggregates in MUT cells were ACTN2-positive (Fig. 3h). Antibodies to various TTN regions (Fig. 3i) revealed that epitopes at the Z/I-junction (2080, T12) adopted a diffusely spread, sparse pattern in MUT cells, suggesting that this TTN region had become uncoupled from the remnant sarcomere-like structures and was rapidly lost (Fig. 3k). A similar pattern of sarcomere disintegration was frequently seen with antibodies to the TTN I/A-junction (I/A-MIR) and M-band (M-TtnK) (Fig. 3l, m). Interestingly, we occasionally observed isolated A-bands in MUT gastrocnemius, which were stained at their edges by I/A-MIR and at their center by M-TtnK (Fig. 3l, m). Such A-bands could appear completely detached from the Z-rods/aggregates, whereas sometimes, the Z-aggregates emerged in the middle of an A-band. These findings confirm that TTN-inactivation causes gradual loss of sarcomere integrity and promotes abnormal protein-protein interactions, including Z-disc-protein aggregation and streaming.

**TTN-based sarcomere disassembly impedes myofiber mechanics**. Next, we sought to test whether loss of TTN affects myofiber mechanical properties. We first measured the lateral stiffness by transverse nanoindentation using atomic force microscopy (Fig. 4a). Single myofibers were dissected from dox-treated WT and MUT (TTN-loss, >80%) muscles, skinned, and attached to glass-bottom dishes coated with Cell-Tak glue in a relaxing buffer (including $Mg^{2+}$-ATP but excluding calcium). Force-indentation curves recorded on myofibers of pectoralis muscle showed significant softening in MUT. The Young's modulus (lateral stiffness) decreased from 7.89 ± 4.65 kPa in WT to 3.92 ± 3.06 kPa in MUT, whereas the indentation depth at 3 nN indentation force increased from 268.5 ± 160.2 nm in WT to 461.4 ± 267.5 nm in MUT (mean ± SD; $n = 119$) (Fig. 4b). Tensile stiffness measured during stepwise stretching of skinned vastus lateralis fibers in relaxing buffer decreased by 75% or more in MUT vs. WT (Fig. 4c). Specific tension of $Ca^{2+}$-activated myofibers was reduced on average by ~70% (Fig. 4d). Thus, the TTN-loss increases myofiber indentability, lowers transversal stiffness, and reduces both passive and active tension.

**TTN depletion causes loss of myosin but not actin and Cronos**. In M. pectoralis from mice treated with doxycycline for at least 4 weeks, full-length TTN protein N2A was reduced in MUT vs. WT by 54 ± 15% (difference between mean ± SEM) relative to

actin protein (Fig. 5a). No consistent changes in actin appeared in MUT and also the NEB/actin ratio was not significantly altered (Fig. 5a). The MyHC/actin ratio was reduced by 45 ± 7%, whereas the TTN/MyHC ratio was unaltered, in MUT vs. WT (Fig. 5a). Similar results were obtained for other fast-type skeletal muscles. Western blotting of M. gastrocnemius using antibodies to MyHC-IIb, the predominant isoform in fast mouse muscles[29], demonstrated downregulation in MUT by 34 ± 12% (Fig. 5b). At the transcriptional level, myosin-heavy-chain-4 (*Myh4*) coding for MyHC-IIb and alpha-1 skeletal-muscle actin (*Acta1*) were not significantly altered in MUT vs. WT, although there was a trend toward reduction in MUT (Fig. 5c). On immunoblots of gastrocnemius or pectoralis muscles, the N2A and Novex-3 isoforms were significantly downregulated in MUT, by 62 ± 23% and 82% (Hodges–Lehmann estimator), respectively (Fig. 5d). Cronos remained unchanged on average, although the expression level varied greatly in MUT.

Similar analyses performed in muscles exposed to doxycycline for 21 days revealed still-preserved N2A/actin and N2A/GAPDH protein ratios in MUT (Supplementary Fig. 6b, c), although N-terminal titin exons were effectively suppressed (Supplementary Fig. 6d); however, Novex-3 protein was already missing (Supplementary Fig. 6c). Cronos TTN was preserved at both protein and transcript levels. Interestingly, the MyHC/actin protein ratio was already reduced in MUT, as was *Myh4* mRNA, whereas actin transcripts (*Acta1*) were unaltered (Supplementary Fig. 6b, d). In summary, loss of Z-disc-anchored TTN caused reduction of myosin but not thin-filament proteins.

**Cronos stabilizes A-bands but does not preserve sarcomeres**. The presence of disconnected A-bands (Fig. 3l, m) and preservation of Cronos expression (Figs. 1e and 5d, Supplementary Fig. 6) in MUT myocytes suggested that remnant A-bands may be stabilized by Cronos. To test this hypothesis, we studied fatigue-resistant skeletal muscles, because we found that in these muscles —specifically, soleus and diaphragm—Cronos protein is not expressed at detectable levels, contrasting the abundant Cronos transcript and protein expression in fast skeletal muscles and the heart (Fig. 6a, b). Immunoblot analysis of TTN in dox-treated WT and MUT diaphragm demonstrated reduction of N2A and Novex-3 in MUT and confirmed the lack of Cronos expression (Fig. 6c, d). Similar to fast-type muscles, the NEB/actin ratio remained unaltered, whereas the MyHC/actin ratio was significantly lower in MUT (1.50 ± 0.10; mean ± SD) than in WT (1.90 ± 0.09) (Fig. 6d), although the relative reduction was less

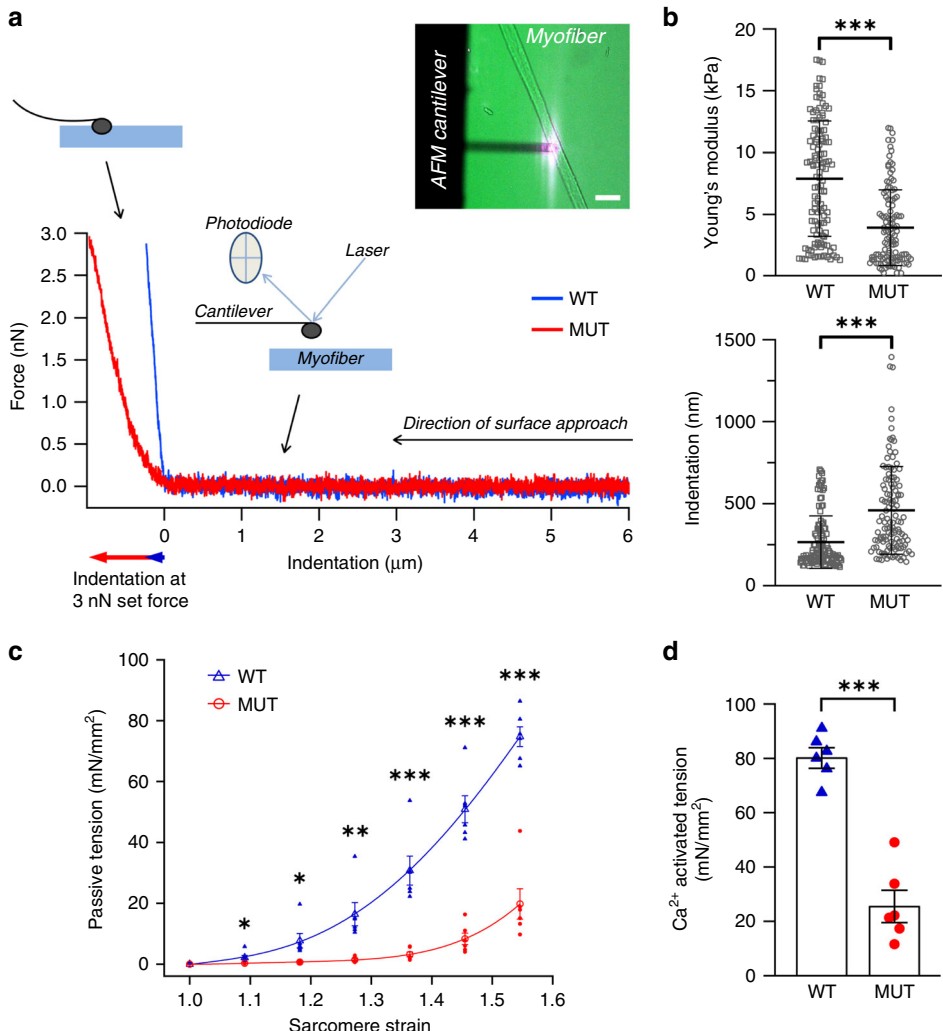

**Fig. 4 Myofiber stiffness and tension from WT and MUT mice after ~30 days of dox-treatment. a** Representative force-indentation curves for myofibers from WT (blue) and MUT (red) mouse pectoralis muscles after dox-treatment. Schematics indicate the principle of measurement by atomic force microscopy (AFM), whereby the cantilever approaches the cell surface until it begins to indent the cell at indentation 0. Indentation was quantified at a pre-set force value of 3 nN. The Young's modulus was calculated by assuming a linear slope of the force-indentation curve at forces above ~0.5 nN. Inset, image of a single myofiber glued to the cover glass with Cell-Tak and indented by the AFM cantilever. Bar, 100 μm. **b** Young's modulus and indentation depth (at 3 nN indentation force) for WT and MUT myofibers treated with doxycycline. $n = 119$ recordings/condition; $n = 6$ m. pectoralis fibers were prepared from $n = 2$ different mice per group. **c** Passive force of isolated, skinned, vastus lateralis WT and MUT myofibers at different stretch states, normalized to cross-sectional area ($n = 6$ fibers from $n = 3$ different mice/group). Strain was estimated from sarcomere-length (SL) measurements by laser diffraction (strain 1.0, slack length of ~2.2 μm SL; strain 1.55, ~3.4 μm SL). Fits are polynomial regressions. **d**) Specific developed tension of vastus lateralis myofibers in activation buffer with $10^{-4.5}$ mol/L free calcium ($n = 6$ fibers from $n = 3$ different mice/group). Skinned myofibers were stretched (strain, 1.2; ~2.6 μm SL) in relaxing solution, held for 2 min, and then maximally activated by calcium. Average data are mean ± SD in **b** and mean ± SEM in **c** and **d**; ***$p < 0.001$, **$p < 0.01$, in unpaired, two-tailed, Student's $t$-test. Source data are provided as a Source Data file.

than in fast-type gastrocnemius muscle (cf., Fig. 5a). Importantly, electron micrographs demonstrated the simultaneous disassembly of all parts of the sarcomere, including the A-band, in MUT diaphragm myocytes (Fig. 6e). While remnant A-bands were never observed, Z-aggregates and Z-disc streaming were frequent. We conclude that Cronos is responsible for transient survival of some A-bands during loss of Z-disc-anchored TTN in fast-type muscles. The lack of Cronos in fatigue-resistant muscles may underlie the general decline of sarcomeric structure in MUT diaphragm. However, even in fast muscles, Cronos at best can delay sarcomere deterioration.

**TTN ablation is accompanied by increased TTN ubiquitination.** In situ hybridization and qPCR measurements demonstrated that very little full-length TTN mRNA remained in MUT muscles after 21–30 days of dox-treatment. Hence, TTN proteins will not be replaced and remaining TTN will eventually be targeted for degradation by the ubiquitination pathway[30]. Analysis of TTN-ubiquitination in fast-type muscle using antibodies to all ubiquitin species (pan-Ub) revealed increased signals for remaining N2A (~3.6 MDa) and Novex-3 (~0.7 MDa) in dox-treated MUT (Fig. 7a). Interestingly, increased ubiquitination in MUT was much more pronounced for the smaller Cronos/T2 TTN species. The proteolytic T2-fragment of TTN (~2.4 MDa) co-migrates with Cronos (~2.3 MDa) on titin gels[31]. No pan-ubiquitination of any TTN species was seen in dox-treated WT. Proteins smaller than TTN were more highly ubiquitinated in MUT than in WT (Fig. 7a). In summary, TTN-inactivation causes increased ubiquitination of remaining TTN protein and other myocyte proteins.

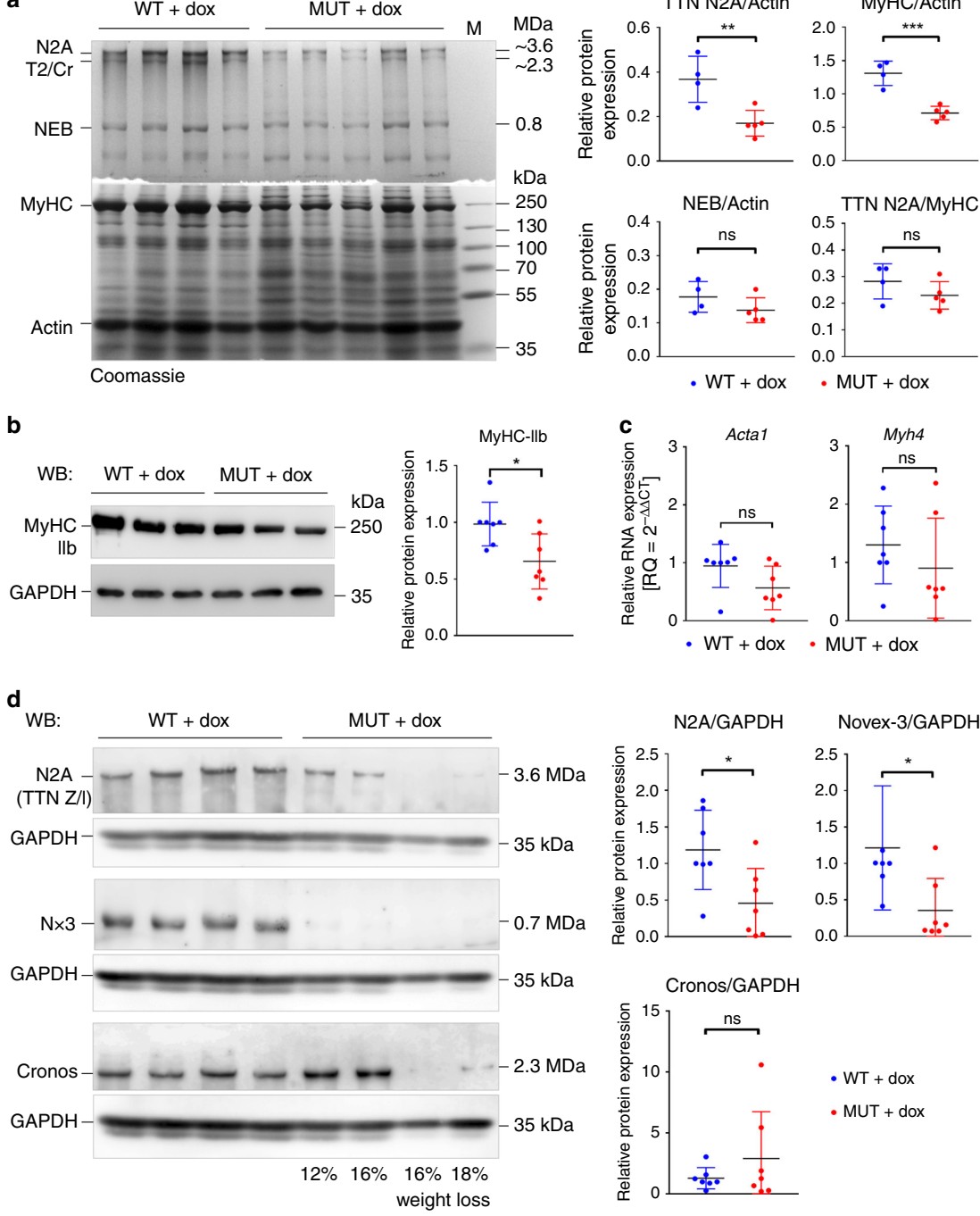

**Fig. 5 Loss of Z-disc anchored TTN reduces myosin but not actin or Cronos. a** Coomassie-stained titin gel (2.5% polyacrylamide) stacked onto a 10% running gel and loaded with protein lysates from dox-treated WT and MUT Ttn^tm1c;ACTA1-rtTA;tetO-cre pectoralis muscle. Right panels show results of densitometric quantification, comparing TTN (N2A), MyHC, or NEB expression to actin levels, as well as TTN (N2A) to MyHC levels, from WT ($n = 4$) and MUT ($n = 5$) mice. NEB, nebulin; MyHC, myosin heavy chain; T2, titin proteolytic fragment; Cr, Cronos isoform; M, marker. **$p = 0.0081$; ***$p = 0.0004$; ns, non-significant. **b** Western blots detecting MyHC-IIb protein in MUT vs. WT gastrocnemius muscles. Right panel shows mean MyHC-IIb:GAPDH ratios in WT (blue) and MUT (red); $n = 7$ mice/group; *$p = 0.0158$. **c** Relative mRNA expression of alpha 1 skeletal muscle actin (*Acta1*) and *Myh4* (codes for MyHC-IIb) in gastrocnemius muscles measured by qPCR; $n = 7$ mice/group; ns, non-significant. N2A titin was probed by antibodies to TTN Z/I (2080). **d** Representative western blots showing expression of TTN isoforms in WT and MUT Ttn^tm1c;ACTA1-rtTA;tetO-cre pectoralis muscle. GAPDH served as loading control. Nx3, Novex-3 isoform. Right panels show results of densitometric quantification of western blots performed on WT and MUT gastrocnemius muscles, indexed to WT levels ($n = 7$ mice/group); *$p_{N2A}=0.0201$; *$p_{Novex-3}=0.0239$. All graphs show mean ± SD. All $p$-values were calculated using two-tailed unpaired Mann–Whitney U-test or two-tailed unpaired Student's t-test. All data were obtained from ~30-day dox-treated muscles. Source data are provided as a Source Data file.

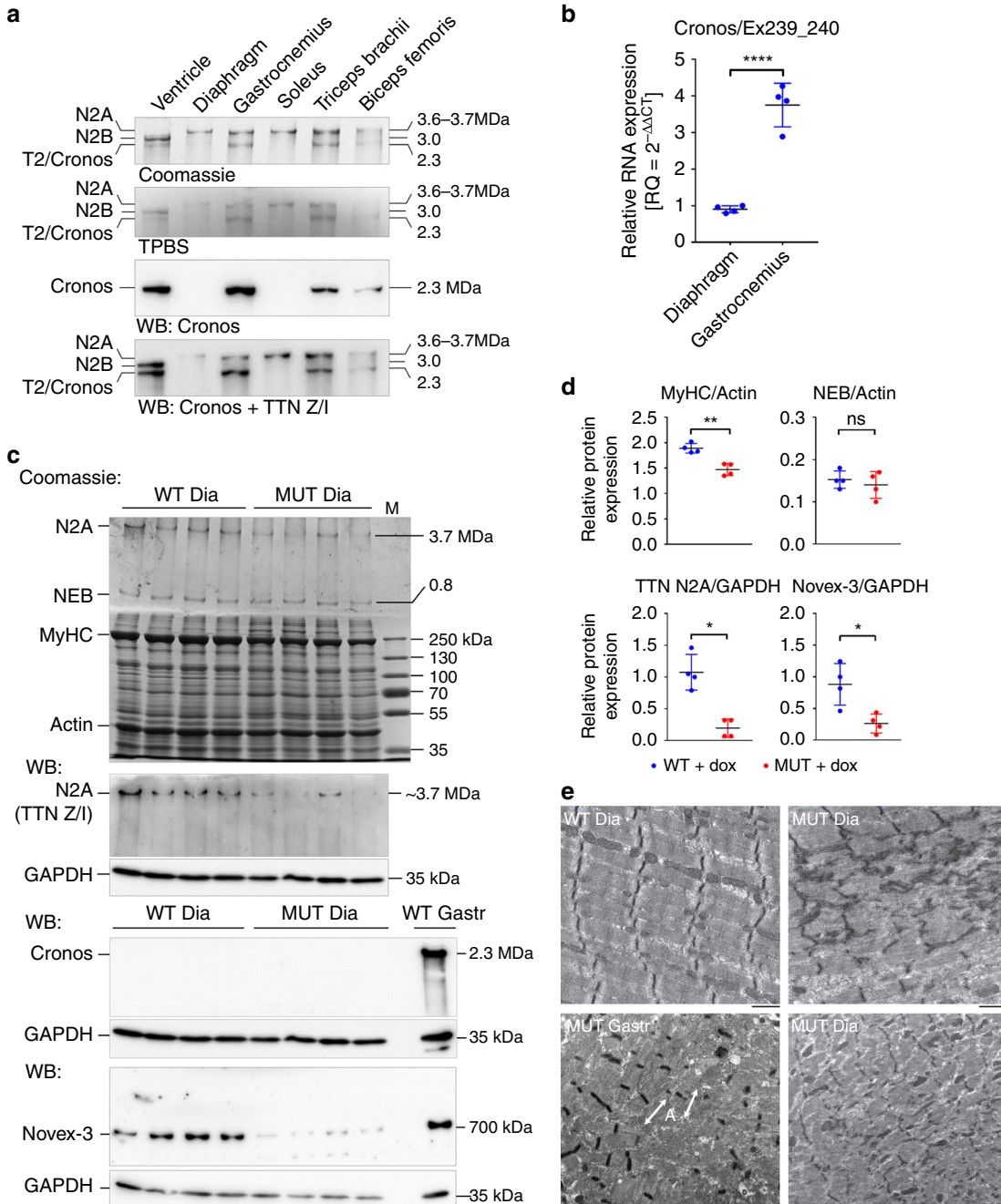

**Fig. 6 Differences in Cronos expression and A-band stability between muscle types. a** Expression of full-length TTN and Cronos in different muscle types of wild-type mice. Top panel, Coomassie-stained, 2% SDS-polyacrylamide gel; lower two panels, western blots (WB) using antibodies to Cronos or both Cronos and full-length TTN (antibody 2080 against TTN Z/I). TPBS, total protein stain of blotted TTN proteins. Gels and blots shown are representative of 2–3 gels/WBs performed per condition. **b** Relative Cronos mRNA expression in wild-type mouse muscles measured by qPCR; $n = 4$ mice/group. ****$p <$ 0.0001, by two-tailed unpaired Student's $t$-test. **c** Coomassie-stained titin gel (2.5% polyacrylamide) stacked onto a 10% running gel and loaded with protein lysates from WT ($n = 4$) and MUT ($n = 4$) Ttn$^{tm1c}$;ACTA1-rtTA;tetO-cre diaphragm muscle after dox-treatment (top panel). M, marker. Lower panels show representative western blots detecting expression of TTN isoforms in dox-treated WT and MUT diaphragm muscles. Full-length titin (N2A) was probed by antibodies to TTN Z/I (2080). GAPDH served as loading control. **d** Results of densitometric quantification ($n = 4$ mice/group); MyHC/ Actin and NEB/Actin ratios were measured on Coomassie-stained titin gel, N2A, and Novex-3 expression on western blots, relative to GAPDH. Data are mean ± SD; **$p = 0.0016$, *$p_{N2A} = 0.0286$, *$p_{Novex3} = 0.014$, ns, non-significant; calculated using a two-tailed unpaired Student's $t$-test. **e** Representative electron micrographs (out of >20 images recorded per condition) of WT (upper left panel) and MUT (right panels) mouse diaphragm muscles after dox-treatment. For comparison, the lower left panel shows a dox-treated MUT gastrocnemius myocyte. Arrows labeled A point to remnant, isolated A-bands. Bars, 1 μm. All data were obtained after ~30 days of dox-treatment. Source data are provided as a Source Data file.

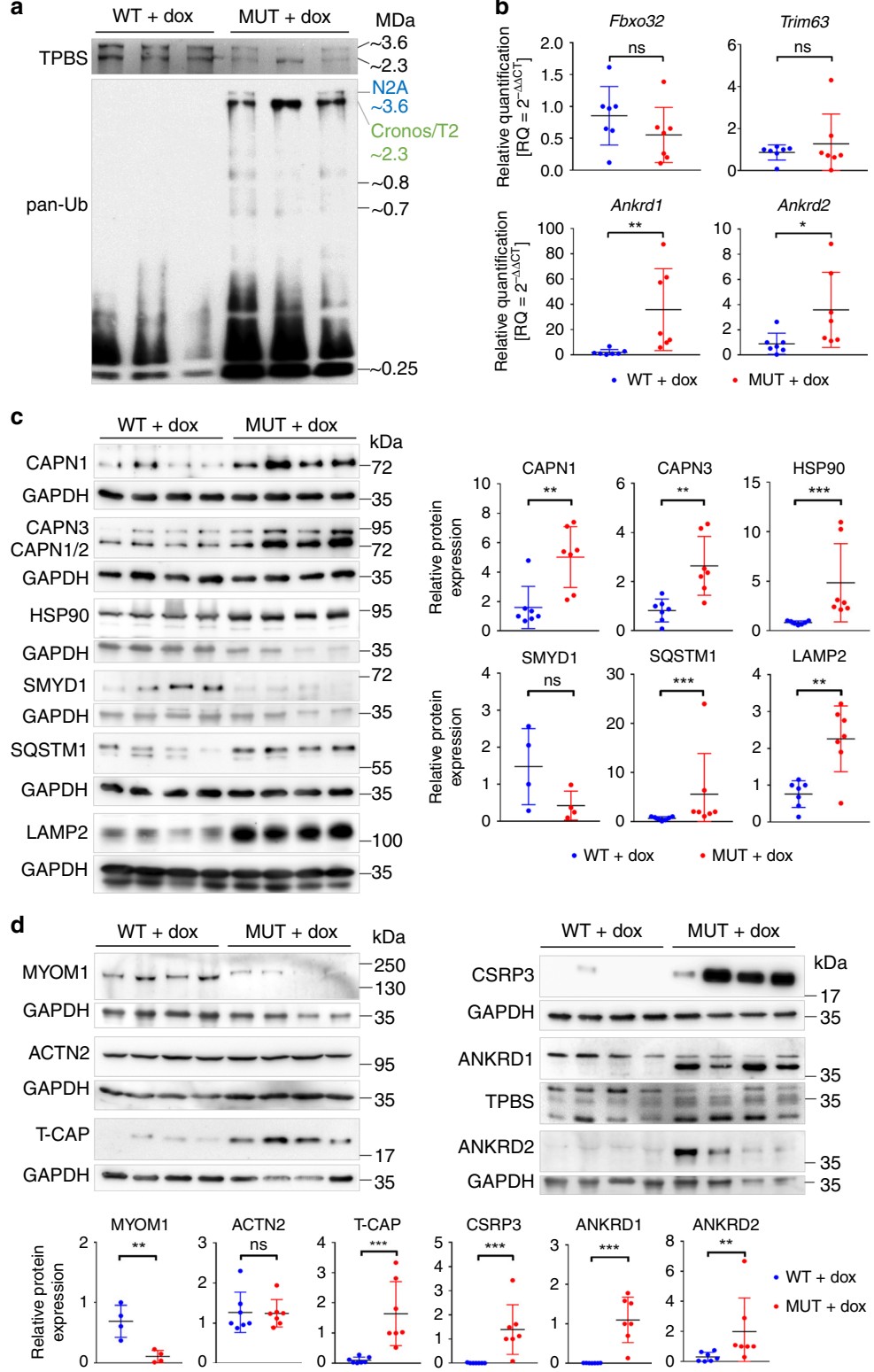

**Loss of TTN activates the PQC machinery**. Sarcomere-bound E3-ubiquitin ligases have been implicated in muscle atrophy, particularly Z-disc-associated *Fbxo32* (MAFbx/atrogin-1) and M-band-associated (TTN-binding) *Trim63* (MuRF1)[32]. However, none of these molecules was significantly altered at transcript level in dox-treated MUT vs. WT M. gastrocnemius (Fig. 7b). In

contrast, two calcium-dependent cysteine proteases, ubiquitously expressed calpain-1 (CAPN1) and skeletal muscle-specific calpain-3 (CAPN3)[33], of which TTN is a known binding partner and substrate, were significantly increased at protein level in MUT vs. WT (Fig. 7c). As expected, anti-CAPN3 antibodies recognized the 94-kDa CAPN3 protein and an additional ~80 kDa band

**Fig. 7 Effects of TTN-loss on PQC markers and TTN-binding mechanosensitive proteins. a** Western blot (2.5% polyacrylamide gel) of gastrocnemius muscle from dox-treated wild-type (WT) and mutant (MUT) mice, using antibodies to pan-ubiquitin (pan-Ub). TPBS, total protein stain of blotted TTN proteins. The blot shown is representative of 2 similar blots performed. **b** Expression of E3 ubiquitin ligases *Fbxo32* and *Trim63* in dox-treated WT and MUT muscles at transcript level by real-time quantitative RT-PCR. In addition, transcript levels for TTN-binding proteins *Ankrd1* and *Ankrd2* are shown. Data are mean ± SD ($n = 7$ mice/group). **c** Expression of PQC markers calpain (CAPN1; CAPN3), heat shock protein 90 (HSP90) and co-chaperone Smyd1, sequestosome-1 (SQSTM1), and lysosome-associated membrane protein 2 (LAMP2) by western blot. Right panels show results of densitometric quantification; $n = 7$ mice/group, except for SMYD1 ($n = 4$). **d** Expression of TTN-binding proteins myomesin-1 (MYOM1), α-actinin (ACTN2), telethonin (T-CAP), muscle LIM protein (CSRP3), Ankrd1 and Ankrd2, by western blot. Lower panels show results of densitometric quantification; $n = 7$ mice/group, except for MYOM1 ($n = 4$). For quantification of signals on western blots, GAPDH served as a reference. ***$p < 0.001$, **$p < 0.01$, ns, non-significant; two-tailed Mann–Whitney test (*Trim63*, *Ankrd1*, CAPN1, HSP90, SQSTM1, LAMP2, ACTN2, T-CAP, CSRP3, ANKRD2) or unpaired, two-tailed Student's *t*-test (*Fbxo32*, *Ankrd2*, CAPN3, SMYD1, MYOM1, ANKRD1). All data were obtained after ~30 days of dox-treatment. Source data are provided as a Source Data file.

corresponding to CAPN1 and/or CAPN2;[34] all these calpain species were upregulated in MUT tissue. Chaperones important for the assembly/maintenance of myosin and TTN were also altered: heat shock protein (HSP)90 was significantly increased in MUT (Fig. 7c), as was the small HSP alpha-β-crystallin (CRYAB; Supplementary Fig. 7), whereas the HSP90 co-chaperone Smyd1 (also important for myosin assembly)[35] showed a strong trend toward reduction in MUT (Fig. 7c). The chaperones HSP70 and HSC70 remained unchanged (Supplementary Fig. 7). Furthermore, the autophagy adapter protein sequestosome-1 (SQSTM1/p62), which also associates with M-band TTN[36], was significantly increased at protein level in MUT vs. WT muscles (Fig. 7c). Likewise, we found significant upregulation of lysosome-associated membrane-protein-2 (LAMP2), which is involved in lysosomal stability and autophagy. In summary, TTN-loss activated crucial factors important for muscle protein degradation and turnover, including relevant chaperones, proteasomal and autophagy-lysosomal pathways.

**TTN-binding mechanosensitive proteins increase with TTN loss.** The high intensity of ACTN2 signals despite the disruption of regular ACTN2 striations in TTN-depleted muscles (Fig. 3) suggested mislocalization/aggregation rather than loss of the Z-disc protein. Indeed, ACTN2 protein expression was not significantly altered in dox-treated MUT vs. WT (Fig. 7d). In contrast, the M-band protein myomesin-1 (MYOM1) was highly reduced in MUT. The N-terminus of TTN complexes not only with ACTN2, but also with telethonin (T-CAP) and cysteine-and-glycine-rich protein-3 (CSRP3; also known as muscle LIM protein, MLP), which has been proposed to act as a mechanosensor[37]. We found that loss of Z-disc-anchored TTN markedly increased T-CAP and CSRP3 protein levels (Fig. 7d). Other titin-binding proteins, four-and-a-half-LIM domain (FHL) 1 and 2[38,39], showed either no significant change (FHL2) or the opposite regulation of two isoforms (FHL1) (Supplementary Fig. 7). Additional molecules associating with I-band TTN include the muscle ankyrin-repeat proteins (MARPs)[40,41]. The expression of ankyrin-repeat-domain-1 (Ankrd1/CARP) and ankyrin-repeat-domain-2 (Ankrd2) was highly increased in dox-treated MUT vs. WT muscles, at both transcript (Fig. 7b) and protein levels (Fig. 7d).

**TTN loss occurs in critical illness myopathy patient muscles.** The progressive loss of thick- but not thin-filament proteins after TTN-inactivation is reminiscent of a special form of human muscle-wasting disease known as CIM, acute quadriplegic myopathy, or myosinopathy[26]. Typically, affected individuals are critically ill ICU patients on long-term mechanical ventilation support, who present with generalized muscle weakness or paralysis due to rapid atrophy of peripheral muscles. Unfortunately, the causes for myosinopathy are poorly understood obfuscating treatment. We confirmed the progressive loss of myosin protein in biopsies of tibialis anterior (TA) muscles from CIM patients (Fig. 8a) and

classified the biopsies into groups with high MyHC:actin (My:Ac) (mean ± SD, 2.10 ± 0.44; $n = 5$ individuals), intermediate (1.39 ± 0.25; $n = 5$), and low expression ratios (0.70 ± 0.34; $n = 4$) (Fig. 8a). The actin-expression levels remained constant even in severely affected CIM muscles (Fig. 8a, Supplementary Fig. 8).

Interestingly, we found strongly reduced TTN:actin ratios in muscles with intermediate- or low-MyHC content, compared to those with high-MyHC content (mean ± SD, 0.71 ± 0.23 (high My:Ac) vs. 0.45 ± 0.15 (intermediate My:Ac) vs. 0.39 ± 0.21 (low My:Ac)) (Fig. 8a, Supplementary Fig. 8). The difference in relative TTN expression was not significant between the intermediate- and low-My:Ac groups, suggesting that most of the TTN-loss occurred at earlier stages of muscle atrophy. As a consequence, the average TTN:MyHC ratio was similar in the high-MyHC (mean ± SD, 0.35 ± 0.14) and intermediate-MyHC groups (0.32 ± 0.06), but significantly increased in the low-MyHC group (0.58 ± 0.23) (Fig. 8a). We also found elevated levels of (pan-)ubiquitinated TTN in CIM muscles with low My:Ac ratios, compared to those with high ratios (Fig. 8b), suggesting increased degradation.

Both the essential and the regulatory myosin light chains were consistently reduced or missing in low-My:Ac CIM muscles (Supplementary Fig. 8a). Another A-band protein, myosin-binding protein-C (MyBPC), was progressively lowered: the MyBPC:actin ratio changed from 0.29 ± 0.07 (high My:Ac; mean ± SD) to 0.19 ± 0.07 (intermediate My:Ac) and 0.09 ± 0.05 (low My:Ac) (Fig. 8a). Furthermore, MYOM1 was nearly absent in CIM muscles with low My:Ac ratios (Fig. 8c). In contrast, nebulin showed little changes with decreasing My:Ac ratios (Fig. 8a). Expression of ACTN2 was identical in muscles with high and intermediate My:Ac ratios, but modestly increased in those with low My:Ac ratio (Fig. 8a). These alterations in sarcomere proteins in CIM patient muscles were accompanied by increased expression levels of other titin-binding molecules in the low-My:Ac vs. high My:Ac groups, which included the chaperone HSP90 (but not HSP70), ANKRD1, and FHL2 (Fig. 8d). Importantly, the changes observed in CIM muscles are (at least qualitatively) highly similar to those in mouse muscles after TTN-inactivation.

Electron microscopy confirmed degradation of thick filaments and preservation of Z-discs and thin filaments in TA muscles from CIM patients with low My:Ac ratios (Fig. 8e). Various changes in the structural organization of myocytes from low-My:Ac-CIM muscles were akin to those observed in TTN-depleted fast-type MUT mouse muscles, including Z-disc deformations and loss of myofibril orientation (cf. Fig. 3g). Taken together, our findings suggest that TTN-loss is a prominent feature of adult muscle remodeling in CIM. In concert with TTN-associated proteins, TTN may act as a key mechanosensor, whose (partial) absence strongly contributes to the rapid muscle atrophy in CIM patients.

## Discussion
Not many details are known about the life cycle of adult muscle sarcomeres, including their maintenance and turnover of proteins. This issue is relevant, because differentiated myocytes are

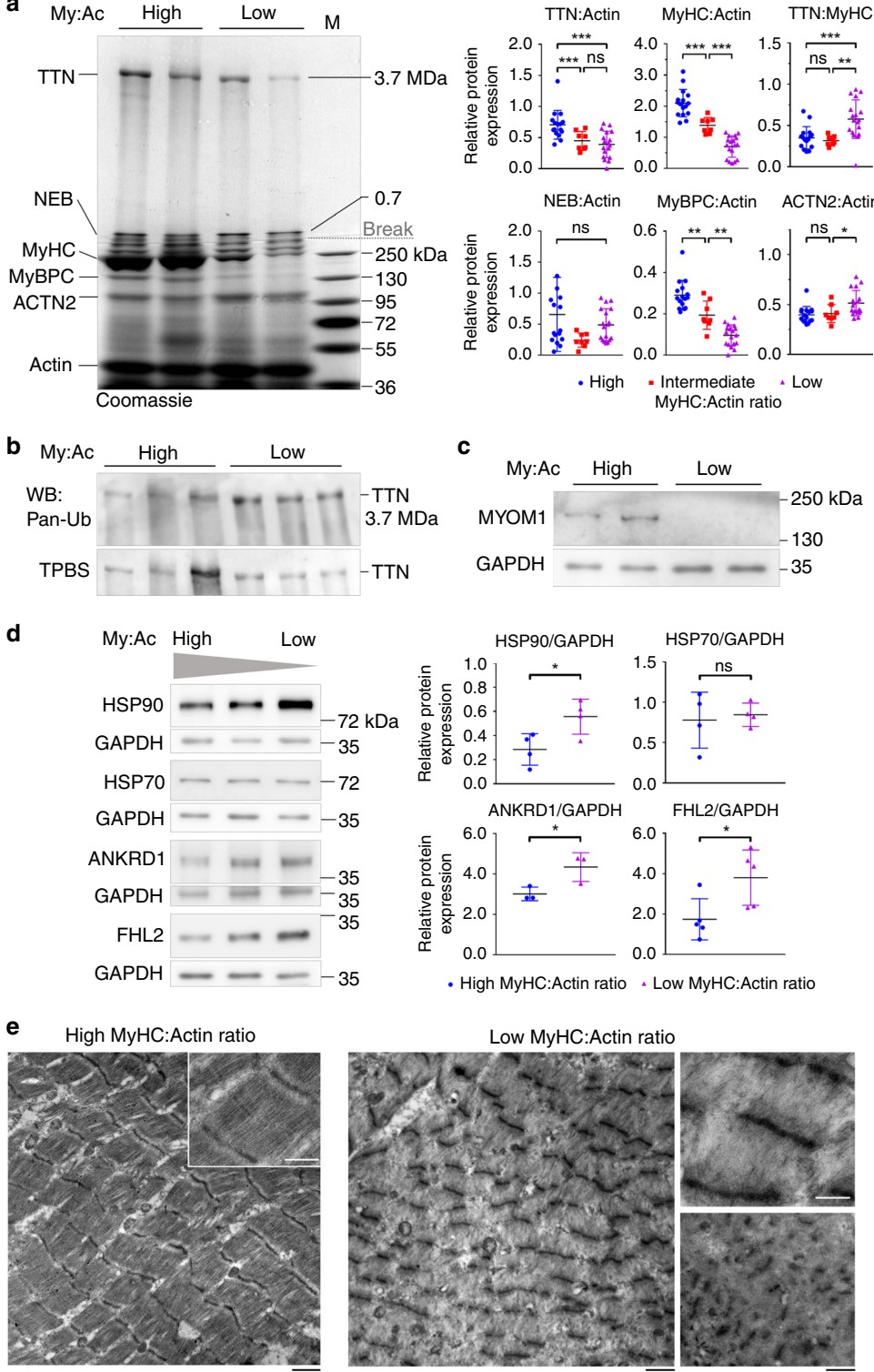

unable to renew and repair themselves by cell division and must continuously replace their building parts, including the sarcomeric proteins, whose lifetime is limited. Since TTN is considered the backbone of the sarcomere, we sought to determine how this protein is involved in the maintenance of sarcomeric integrity by removing the Z-disc-anchored isoforms in adult murine skeletal muscle. We observed a dramatic loss of muscle function beginning ~3 weeks after the supply of new TTN stopped, concomitant

with the gradual disintegration of the sarcomeres, which is consistent with a TTN half-life of 2–3 weeks[42]. TTN-loss caused reduction of thick- but not thin-filament proteins, myocyte mechanical impairment, activation of PQC systems, and increased expression of TTN-binding mechanosensory proteins. These findings clearly indicate that Z-disc-anchored TTN is crucial for sarcomere stability and function in adult muscle. Remarkably, we find that progressive loss of full-length TTN

**Fig. 8 Protein expression and ultrastructure of CIM patient muscles. a** Left panel shows a representative Coomassie-stained titin gel (1.8% polyacrylamide) stacked onto a 6% running gel and loaded with biopsy samples of tibialis anterior muscles from critical illness myopathy (CIM) patients expressing high or low myosin heavy chain (MyHC) over actin (My:Ac) ratios. Right panels show results of densitometric analysis of Coomassie-stained stacking gels. Quantification of relative sarcomeric protein expression was done separately for CIM muscles with high ($n = 5$ patients), intermediate ($n = 5$ patients), and low ($n = 4$ patients) My:Ac ratios. Independent gel electrophoretic analysis was done three to four times/individual for the high-My:Ac and low-My:Ac groups, and usually two times/individual for the intermediate-My:Ac group; all data points are shown. **b** Western blot of 1.8% polyacrylamide gel loaded with biopsy samples of CIM muscles expressing high or low My:Ac ratios, using antibodies to pan-ubiquitin (pan-Ub). TPBS, total protein stain of blotted TTN proteins. **c** Western blot detecting expression of myomesin-1 (MYOM1) in CIM muscles with high or low My:Ac ratios. GAPDH, loading control. Blots shown in **b** and **c** are representative of three total blots performed. **d** Western blots detecting levels of heat shock proteins (HSP90, HSP70) and titin-binding mechanosensitive proteins (ANKRD1, FHL2) in CIM muscles expressing high to low My:Ac ratios. GAPDH, loading control. Right panels show results of densitometric analysis. Analysis was done on $n = 4$ patients/group for HSP90 and HSP70; $n = 3$ patients/group for ANKRD1, and $n = 5$ patients/group for FHL2. **e** Representative electron micrographs (out of >25 images recorded per condition) of CIM muscles expressing high (left panel) and low My:Ac ratios (right panels). Note the pattern of sarcomere disassembly with preferential thick-filament depletion, loss of sarcomere orientation, and Z-disc protein aggregation. Bars, 1 µm (black bars), and 0.5 µm (white bars). Data in **a** and **d** are mean ± SD. \*\*\*$p < 0.001$, \*\*$p < 0.01$, \*$p < 0.05$, ns non-significant; unpaired, two-tailed Student's $t$-test with Welch's correction; for multiple comparisons, ANOVA followed by Sidak's test was also performed. Source data are provided as a Source Data file.

protein is characteristic of skeletal muscles from ventilated CIM patients in ICUs and that the TTN-loss accompanies the well-known preferential loss of thick filaments in CIM. We propose that the sarcomere disruption and muscle wasting in peripheral skeletal muscles of CIM patients are, at least in part, due to mechanical silencing of titin.

Interestingly, our proof-of-concept homozygous TTN-KO mice failed to develop aligned cardiac sarcomeres. This finding is consistent with previous reports of genetically modified mice[14] and human-induced pluripotent stem cell-derived cardiomyocytes (hiPSC-CMs) carrying TTN-truncating mutations, which suggested that contiguous TTN is critical for the initial stages of sarcomere assembly[43]. Further analysis of hiPSC-CMs with an A-band-TTN truncation implicated that intact TTN forms an essential mechanical link between sarcomeric myosin and the protocostamere-actinin (Z-body) complex as a prerequisite for sarcomere assembly[13]. Our constitutive TTN-KO model supports the view that full-length TTN provides a mechanical connection between the Z-bodies and the myosin filaments required for cardiac sarcomerogenesis. The A-band-bound Cronos isoform, which we recently found to support formation of sarcomeres in hiPSC-CMs[31], does not rescue cardiac sarcomerogenesis in vivo in the absence of Z-disc-anchored titin.

Although TTN is widely thought to stabilize adult sarcomeres, direct proof has been missing. Our inducible, skeletal muscle-specific TTN-KO mouse now unequivocally demonstrates that Z-disc-anchored TTN maintains the sarcomeric integrity in mature muscle cells. While this function is provided most likely by the half-sarcomere-spanning titin isoforms (such as N2A), it remains to be seen whether Novex-3, which was also deleted in our model, has an additional role in sarcomeric maintenance.

We found differential expression of Cronos in fast-type vs. fatigue-resistant murine skeletal muscles and exploited this fact to address the potential role of Cronos in rescuing the sarcomere structure[5] after depletion of Z-disc-anchored TTN. However, disintegration of the sarcomeres occurred independent of whether Cronos was present (M. gastrocnemius) or not (diaphragm), which excludes a role of Cronos in maintaining the structure of adult sarcomeres. Remarkably, there was a tendency for extended lifespan of remnant A-bands in muscles expressing Cronos. Overall, however, the protective effect of Cronos on the A-bands, if any, appeared to be rare and transient, and it did not prevent rapid detachment of Z-discs from A-bands upon TTN-KO. Generally, the changes in sarcomeric protein levels and ultra-structure with TTN-loss were similar in all muscle types studied. Thus, Cronos may play a role in stabilizing the A-band of healthy

fast-type muscle sarcomeres, but it is unable to substitute for the function of full-length TTN in sarcomere proteostasis.

The role of TTN as a contributor to myocyte mechanical properties was confirmed here, as both passive stiffness (lateral and longitudinal) and active developed tension were reduced in dox-treated MUT vs. WT myofibers. An additional property of TTN is its mechanosensor function[44–46]. Mechanosensitive regions in titin's Z-, I- and M-band segments associate with molecules involved in hypertrophic/atrophic signaling, which are activated by stress or strain[47]. In the Z-disc, the TTN N-terminus interacts with the T-CAP/CSRP3 complex, a postulated key component of Z-disc-based mechanosignaling[37]. We found that TTN-loss in adult muscle profoundly increased T-CAP and CSRP3 expression. Previously, transgenic mice overexpressing T-CAP were reported to develop less apoptosis than WT animals under elevated cardiac afterload, suggesting a role for the protein in regulated cell death[48]. CSRP3 is involved in myocyte autophagy, next to its roles in myocyte cytoarchitecture and myotube differentiation[49]. Considering these functions, the upregulation of T-CAP and CSRP3 in our MUT mouse may represent an adaptive mechanism to prevent muscle wasting—which however fails, as TTN becomes increasingly unable to provide mechanical support.

Other mechanical signaling molecules in myocytes binding to the N2A-region of I-band TTN are the MARPs[40]. Members of this protein family can modulate transcription factors and are deregulated under various muscle stresses[50]. The increase of Ankrd1 and Ankrd2 in our TTN-KO mice suggests deregulation of hypertrophy/atrophy signaling. Furthermore, the N2A-element and many Ig-domains in the TTN spring interact under stress with small HSPs, such as CRYAB[51]. Chaperones are important regulators of myocyte protein turnover and provide a first-line protective response to many types of stress[52]. At least part of the increase in CRYAB expression observed upon TTN-inactivation may be related to disrupted sHSP-titin interactions. The N2A-element of TTN also associates with HSP90 in complex with co-chaperone Smyd2; together, they protect the Z/I-band structure of the sarcomere[53,54]. HSP90 and another Smyd protein, Smyd1, also have crucial roles in chaperoning the myosin filament during sarcomere development and maintenance[35,55]. The abnormal expression of these (co)chaperones in our MUT muscles may indicate the (failed) attempt to protect sarcomeric proteins from pathological aggregation and excessive degradation.

Various other components of the cellular PQC machinery became activated as a consequence of the TTN-loss. The TTN-binding calpains, CAPN1 and CAPN3[56,57], were both elevated.

Their increase suggests a role in sarcomeric protein degradation at early stages of myofibril breakdown[33,58] in our MUT model. Furthermore, we observed substantially higher ubiquitination of TTN and other muscle proteins in dox-treated MUT vs. WT muscles as a sign of increased activity of the ubiquitin-proteasome system and alternative pathways for targeted degradation[30,52]. This finding is consistent with previous studies in mouse models of muscle atrophy demonstrating that TTN becomes ubiquitinated at lysine residues mainly in the A-/M-band[59,60]. Interestingly, the muscle E3 ubiquitin ligases, *Fbxo32* (atrogin-1) and *Trim63* (MuRF1), were not significantly altered in our TTN-KO mice, although they are usually induced during muscle atrophy[32]. While TRIM63 binds directly to TTN at the M-band[61], the E3 ligase has never been shown to ubiquitinate TTN; however, it does ubiquitinate many other sarcomere proteins, preferentially in thick filaments[62]. We speculate that other, yet-to-be-identified, E3 ligases may be activated by reduced supply of TTN, causing increased ubiquitination of remaining TTN protein. Moreover, two markers of autophagic-lysosomal activity, SQSTM1/p62 and LAMP2[63,64], were upregulated in MUT muscles. SQSTM1 binds via neighbor of BRCA1 gene-1 protein to the titin-kinase domain in M-band TTN[36], thus linking stress-induced autophagic signaling to the sarcomere[47]. Although it is still unknown whether/how much TTN protein is degraded via autophagic pathways, the increase in autophagy markers upon TTN-KO suggests involvement of autophagic processes in the sarcomeric disintegration. Collectively, these results demonstrate that the TTN-loss increases the activity of all components of the PQC machinery, which jointly mediate the accelerated degradation of the myocyte proteins.

An intriguing finding of this study is the remarkable similarity in the pattern of sarcomere disintegration and myocyte protein changes between our TTN-KO mice and CIM patients presenting with peripheral muscle wasting. In TA muscles of CIM patients, progressive reduction in thick-filament proteins, such as MyHC, MyBPC and myomesin-1, was accompanied by a significant loss of TTN protein already at early stages of the muscle atrophy when the MyHC:actin expression ratio was only modestly reduced. Nebulin, actin, and α-actinin levels showed little changes, whereas TTN-binding mechanosensitive proteins, chaperone HSP90, and the levels of ubiquitinated TTN increased progressively. A large body of evidence has demonstrated that CIM patient muscles upregulate, among others, components of the PQC machinery, including chaperones, proteases, and E3-ubiquitin ligases[65]. *Trim63* activation was speculated to cause the preferential myosin-loss in CIM patients[26]. Other proteins found to be elevated in CIM patient muscles are inflammatory molecules, transcriptional repressors, and mechanosensitive myocyte proteins[65]. We are not aware of any genetic animal model resembling the hallmarks of human CIM, i.e., the preferential loss of thick filaments and the pathophenotypic appearance of the myocyte/sarcomere ultrastructure, as closely as induced depletion of TTN.

In mechanically ventilated, deeply sedated and/or pharmacologically paralyzed ICU patients, there is a complete loss of mechanical stimuli of the skeletal muscles, including external strain (no weight bearing!) and internal strain (no activation of the contractile apparatus!), which presumably is an important trigger of this special myopathy. A sophisticated rat model has been developed to assist in defining the contributions of mechanical ventilation and muscle unloading to CIM[66,67]. Importantly, passive mechanical loading attenuated muscle wasting and myosin loss in this model[68]. Thus, mechanosensitive factors within the muscles are involved in the pathomechanisms of CIM. Our findings suggest the participation of TTN-based mechanosensing via TTN-binding signaling molecules.

Mechanical silencing of TTN due to muscle unloading may cause the loss of TTN in CIM, because mechanosensitive pathways normally triggered by TTN strain fail to be activated and regulate sarcomere protein turnover. Despite upregulation of TTN-associated factors, muscle wasting proceeds, and the pathologically lowered TTN levels cause sarcomeric disintegration. Presumably, loss of TTN precedes loss of myosin not only in our TTN-depletion mouse model but also in CIM. If this holds true, the TTN-KO mouse provides a new tool for studying the pathogenesis of CIM and may also help in the search for potential therapeutic approaches.

## Methods

**Animal experiments.** All animal procedures were performed according to national guidelines (TierSchG and TierSchVersV) and were approved by local government authorities (Landesamt für Natur, Umwelt und Verbraucherschutz Nordrhein-Westfalen, LANUV, Az 84-02.04.2015.A261).

**Animal models.** The titin knockout-first mice were generated by injecting EPD0145_2_G01 ES cell clones from the EUCOMM (European Conditional Mouse Mutagenesis Program) Consortium into BAlb/C blastocysts. The specifics and the complete DNA sequence of the *Ttn* gene targeting construct are available in the GenBank® database (accession no. JN959336.1). Validation of inserted cassette integrity and correct targeting are described in supplementary methods (Supplementary Fig. 1). The chimeric mice obtained were mated to C57BL/6JRj mice (Janvier Labs, France) to generate heterozygous knockout-first mice (Ttn^tm1a/+) and to Flipase (Flp)-expressing transgenic mice (Flp-Deleter)[69] to generate the lacZ-neo cassette-deleted, heterozygous, conditional Ttn KO mice (Ttn^tm1c/+).

The inducible skeletal muscle-specific Ttn KO mice were generated by crossing the conditional Ttn KO mice (Ttn^tm1c/+ or Ttn^tm1c/tm1c) to the B6;C3-Tg(ACTA1-rtTA,tetO-cre)^102Monk/J mouse line, purchased from the Jackson Laboratory (Bar Harbor, Maine, USA), which expresses Cre-recombinase under the control of the tetracycline-responsive regulatory element (tetO/TRE) and a reverse tetracycline-controlled transactivator (rtTA) under the control of the human actin, alpha 1 skeletal muscle (*ACTA1*) promoter[28].

As a reporter line for the effectiveness of the dox-treatment, we used a Rosa26-LacZ/Tg(ACTA1-rtTA,tetO-cre) hybrid line, generated by mating the B6;C3-Tg (ACTA1-rtTA,tetO-cre)^102Monk/J mouse line to the B6;129S4-Gt(ROSA) 26Sor^tm1Sor/J strain. The latter is the R26R Gtrosa26 targeted mutant mouse[70] carrying lacZ flanked by loxP sites (Supplementary Fig. 4).

Primers used for genotyping are shown in Supplementary Table 1.

**Transgene induction.** In 8–30 week-old Ttn^tm1c/tm1c;ACTA1-rtTA^pos;tetO-cre^pos and gender-matched WT littermates, Cre-mediated recombination was achieved by administering the tetracycline analog doxycycline. Doxycycline hydrochloride (dox; Sigma, D9891) was dissolved in drinking water at a concentration of 2 mg/mL supplemented with 5% sucrose and was provided ad libitum in light-proofed water bottles for 27–38 days. Treatment was terminated when mice had reduced their body weight by 10–20%. Tissues were then collected from mutant and WT mice for further experiments.

**Morphometric and histological analyses.** Mice were sacrificed by cervical dislocation and embryos, adult hearts and skeletal muscles were dissected. Body weight was recorded and muscle weights measured relative to tibia length. Tissues were fixed with 4% paraformaldehyde (PFA) in phosphate-buffered saline (PBS) for paraffin- or cryo-embedding, or with Zamboni's fixative for electron microscopy, and placed on a shaking table at 4 °C overnight. Cryosections of M. triceps brachii were stained with hematoxylin or hematoxylin/eosin to observe differences between dox-treated WT and MUT. Digital images of hematoxylin/eosin-stained cryosections of triceps brachii were processed and analyzed using ImageJ software (NIH, Bethesda, USA), to measure cell number, fiber cross-sectional area, and number of nuclei. The percentage of centrally nucleated fibers was determined for at least 225 (up to 488) fibers per muscle, using photomicrographs taken at two different positions along the muscle. The cross-sectional area was determined on at least 160 fibers per animal at different positions along the entire muscle.

**Muscle physiology assessment.** The 4LHT or inverted screen test[71] was used to test muscle strength of dox-treated Ttn^tm1c/tm1c;ACTA1-rtTA^pos;tetO-cre^pos and WT mice. Briefly, mice were placed on a wire mesh consisting of 6 mm squares of 1 mm diameter wire, surrounded by 5 cm edging to prevent climbing on the other side. Immediately after placing the mouse in the center, the screen was turned upside down, fixed in this position 40–50 cm above a soft-bedded cage, and the hang time was measured. The test ended when a hang time of 15 min was reached; if not, the test ended after two sessions. The maximum hang time was recorded.

**In situ hybridizations**. Following overnight-fixation in 4% PFA, tissues were dehydrated in a graded ethanol series, paraffin-embedded, cut into 8 μm sections, and mounted on Superfrost™ Plus microscope slides. The *Ttn* exon 20_24 probe was derived from *Ttn* N2A sequence positions 3372–3483 (reference sequence: NM_011652.3, primer sequences in Supplementary Table 1) and digoxigenin-labeled according to the manufacturer's specifications (Roche, Basel). In situ hybridizations on sections were performed[72] with the following modifications: sections were pre-hybridized in hybridization mix without probe for 1–2 h at 60 °C and then hybridized overnight at 60 °C. Approximately 250 μL of hybridization mix were applied to the slide and the sections covered by a coverslip. Probe concentration was ~1 ng/μL.

**RT-PCR and quantitative real-time PCR analysis**. Total RNA was prepared from tissues using Trizol® (Invitrogen) according to the manufacturer's protocol. RNA quality and quantity were estimated by comparing the fluorescence signals of the 28 S and 18 S rRNA bands on agarose gels. Reverse transcription was carried out using Oligo(dT) and Random primers, and RNA polymerase M-MLV RT (Promega, Madison, USA) according to the manufacturer's instructions. Transcript-specific primers are reported in Supplementary Table 1.

Quantitative real-time PCR (qPCR) analysis was performed on samples obtained from at least 3 animals per genotype and used for quantification of relative gene-expression data using the RQ ($2^{-\Delta\Delta CT}$). The hypoxanthine-guanine phosphoribosyl transferase gene (*Hprt*; accession, NM_013556) served as the reference gene, unless otherwise stated. The qPCR was performed on the StepOne™ Real-Time PCR System (Applied Biosystems) using the following 2-step program: initial denaturation at 95 °C for 10 min followed by 40 cycles of denaturation for 15 s at 95 °C and annealing/extension for 1 min at 60 °C, terminated by a final denaturation step at 95 °C for 15 s. For qRT-PCR reactions, the GoTaq qPCR Master Mix reagent was used. Amplicon specificity was confirmed by examination of the dissociation curves. A distinct single peak indicated that a specific DNA sequence had been amplified.

**SDS-PAGE and western blot analysis**. Snap-frozen heart and skeletal muscle tissues were solubilized in titin sample buffer (8 M urea, 2 M thiourea, 3% SDS, 0.05 M Tris, 10% glycerol, 75 mM DTT, 0.03% Serva Blue) by cutting them in very small pieces and incubating them for 20 min on ice. After boiling the probes for 3 min at 96 °C, the debris was spun down for 5 min at 20,000 × g and directly loaded on gels or frozen at −80 °C until use.

The concentration of total protein in each sample was assessed using a Bradford reagent (Sigma). For the separation of giant myofibrillar proteins, protein lysates (7–30 μg) were resolved overnight in agarose-stabilized 1.8–3% slab gels[23,34]. Proteins up to 200 kDa were resolved by standard SDS-PAGE (15%, 12.5%, 10%, or 8%). For simultaneous separation of TTN and proteins up to 250 kDa, we combined both methods, using for the stacking gel an agarose-stabilized 1.8–2.5% gel and for the running gel the standard SDS-PAGE recipe (12.5%, 10%, 8%, 6%). Electrophoresis was performed at 2 mA/plate at room temperature overnight. After separation, proteins were transferred onto PVDF membranes (Millipore) by using the Trans-Blot® Turbo™ transfer system (Bio-Rad). After transfer, total protein blot stain (TPBS) on the membranes was performed with Coomassie brilliant blue (1610436, Sigma-Aldrich). Myosin or actin bands were used as an additional loading control.

Detailed information on the primary and secondary antibodies is provided in Supplementary Table 2. Validation of custom-made antibodies is described in Supplementary Methods and results are shown in Supplementary Fig. 9. Signals from HRP-conjugated secondary antibodies were visualized by using chemiluminescence (Amersham ECL start Western Blotting Detection Reagent, GE Healthcare) and recorded using the ImageQuant LAS 4000 Imaging System (GE Healthcare). Signal intensity was quantified using the ImageQuant TL software (GE Healthcare).

**Immunohistochemistry**. Muscle tissues were prepared for immunohistochemistry using standard protocols. Briefly, samples were fixed with 4% formaldehyde solution in PBS and embedded in paraffin; 10-μm-thick sections were prepared. After deparaffinization and rehydration, antigen unmasking was achieved by shortly boiling the slides, submersed in 1 mM EDTA (pH 8), using a microwave oven. After incubation for an additional 20 min at a sub-boiling temperature, slides were washed in distilled water and incubated in blocking buffer (0.1% Triton X-100 and 10% goat serum in PBS) for 1 h. Antibodies were diluted in staining buffer (0.1% bovine serum albumin in PBS). The primary antibodies were applied overnight at 4 °C, the secondary antibodies for 2 h at room temperature, with three washes in between incubations. Detailed information on the primary and secondary antibodies is given in Supplementary Table 2. Sections were embedded in Mowiol and images were recorded at room temperature using a confocal laser scanning system (Nikon A1) equipped with an Eclipse Ti inverted microscope.

**Quantification of cross striations**. Immunofluorescence staining on paraffin sections was performed using anti-titin Z-disc (TTN Z/I-2080) and anti-α-actinin antibodies (Supplementary Table 2) to determine the number of cells with or without cross striations in WT and MUT muscles (*n* = 3 gastrocnemius muscles

per group) after dox-treatment for 21 or ~30 days. Images were recorded using a Nikon A1 confocal microscope. The striation pattern of at least 90 muscle fibers per animal was evaluated.

**Cryo embedding**. After incubation in 4% PFA at 4 °C overnight, muscle tissues were placed in 15% sucrose in PBS, then in 30% sucrose in PBS and slightly agitated at 4 °C, until the tissue had receded. Afterward, tissues were embedded in optimal cutting temperature compound and cooled down on dry ice. Cryo-sectioning was performed on a Leica CM300 Cryostat.

**Electron microscopy and immunoelectron microscopy**. Mouse skeletal muscle and CIM patient biopsy samples were fixed in 4% PFA, 15% saturated picric acid in 0.1 M PBS, pH 7.4, at 4 °C overnight. For ultrastructural analysis without antibody labeling, tissue samples were embedded in resin and processed for transmission-EM. For immuno-EM, longitudinal sections were cut on a vibratome (Leica VT 1000 S) at a thickness of 50 μm, blocked in 20% normal goat serum (NGS, Vector Laboratories, Burlingame, USA) in PBS and were incubated with primary antibodies diluted in PBS containing 5% NGS overnight at 4 °C. Antibody references and corresponding dilutions are provided in Supplementary Table 2. After several washes in PBS, the sections were incubated with 1.4 nm gold-coupled secondary antibodies (Nanoprobes, Yaphank, NY, USA) in PBS at 4 °C overnight. After extensive washing, the sections were postfixed in 1% glutaraldehyde in PBS for 10 min and then reacted with HQ Silver kit (Nanoprobes). Following treatment with osmium tetroxide, sections were stained with uranyl acetate, dehydrated, and embedded in Durcupan resin (Fluka, Switzerland). Ultrathin sections were prepared (Ultracut S; Leica, Germany) and adsorbed onto glow-discharged Formvar-carbon-coated copper grids. Microscopy was performed on a Zeiss LEO 910 electron microscope and images were taken with a TRS sharpeye CCD camera (TRS Systems, Moorenweis, Germany).

**Quantification of Z-disc perimeter**. On electron micrographs of WT and KO (10–20% BWL) ultrathin sections of gastrocnemius muscle tissue (*N* = 2 animals, 250 Z-discs/group), we calculated the Z-disc circumference by using ImageSP software (TRS Systems, Moorenweis, Germany). Z-disc boundaries were clearly identified by their electron densities and perimeters were obtained via ImageSP polygon measurement. Full image information was transferred to Microsoft Excel for statistical analysis.

**AFM-based myofiber stiffness measurements**. To determine the mechanical stiffness of WT and MUT skeletal muscle cells, single isolated fibers were probed by nanoindentation measurements using an atomic force microscope (Nanowizard 3 AFM system, JPK, Germany). Deep-frozen tissue was defrosted and skinned overnight in ice-cold low ionic-strength buffer (75 mM KCl, 10 mM Tris, 2 mM MgCl₂, 2 mM EGTA, and 40 μg/mL protease inhibitor leupeptin, pH 7.2) supplemented with 0.5% Triton X-100. Under a binocular (Leica, Mannheim, Germany), single myofibers were manually dissected and placed in relaxing buffer (8 mM ATP, 20 mM imidazole, 4 mM EGTA, 12 mM Mg-propionate, 97 mM K-propionate, pH 7.2) at room temperature. The nanoindentation technique was applied as follows: dissected myofibers in relaxing buffer were placed in a fluid chamber, in which the cover glass had been coated with the glue Cell-Tak (DLW354240, Sigma-Aldrich, Germany); fibers thus adhered tightly to the cover glass as a prerequisite for reliable force-indentation measurements. A pre-calibrated cantilever (spring constant, $k_C = 0.03$–$0.04\,N\,m^{-1}$, Novascan Technologies, USA) with a mounted spherical polystyrene bead (diameter, 10 μm) periodically indented the myofiber up to a pre-set indentation force of 3 nN. During the measurements, the AFM tip was moved vertically toward the sample surface at constant velocity of $2\,\mu m\,s^{-1}$. Upon indentation of the myofiber, the AFM cantilever became deflected. The relative position of a laser beam, focused on the cantilever tip and reflected back to a photodiode detector, was used to quantify cantilever deflection and generate force-vs.-indentation curves. From the force acting on the cantilever, the piezo displacement, and the deflection sensitivity, the indentation depth was calculated. By subtracting the deflection of the cantilever from the piezo movement after contacting the myofiber, the force-distance curves were converted to force-indentation curves. The slope of the force-indentation curve indicated the lateral stiffness of the myofiber. The Young's modulus was obtained by fitting the force-indentation curve from the contact point to the end (maximum indentation) using the Hertz model corrected for a spherical indenter (JPK data processing software, JPK, Germany), according to the equation:

$$F = \frac{4E \cdot R^{\frac{1}{2}}}{3(1 - \nu^2)} \cdot \delta^{\frac{3}{2}} \qquad (1)$$

where $F$ is the indentation force, $E$ is the Young's modulus, $R$ is the radius of the sphere directly contacting the sample surface ($R = 5$ μm in the current experiments), $\nu$ is the Poisson's ratio of the sample ($\nu = 0.5$, commonly used for biological samples), and $\delta$ is the measured indentation depth.

**Passive and active tension measurements on single myofibers**. Dissected vastus lateralis muscle from ~30-day dox-treated WT and MUT mice was skinned

overnight in ice-cold relaxing buffer (2.5 mM ATP, 14.5 mM phosphocreatine, 20 mM MOPS, 5 mM K$_2$EGTA, 2.5 mM MgAcetate, 170 mM K-propionate, pH 7.2) supplemented with 0.5% Triton X-100 and protease inhibitor cocktail (G6521, Promega). After washing three times with relaxing buffer, myofiber isolation was carefully done on ice, avoiding excessive stretching. Single myofibers were attached between a motor and a force transducer (Scientific Instruments, Heidelberg, Germany) and mechanical experiments run at room temperature. Fiber cross-sectional area was estimated from the fiber diameter (assuming a circular shape) measured at slack length. For passive tension measurements, fibers were stretched from slack length in six equidistant steps (maximum strain, 1.55). After each step, force relaxation was allowed for during a 1-min hold period. The peak force value immediately after each stretch was converted to stress (force per cross-sectional area). For active tension measurements, fibers were pre-stretched (strain, 1.2) in relaxing buffer for 2 min and then transferred to activating solution (pCa 4.5; 2.5 mM ATP, 10 mM MOPS, 5 mM CaEGTA, 2.4 mM Mg-Acetate, 170 mM K-propionate, pH 7.2), while force was recorded. Maximum active force was normalized to fiber cross-sectional area (Ca$^{2+}$-activated tension). All solutions used for mechanical experiments contained protease inhibitor cocktail (G6521, Promega).

**Biopsies of CIM patient muscles**. Percutaneous conchotome muscle biopsies were taken from the *TA* muscle of 9 mechanically ventilated ICU patients with CIM (intermediate myosin:actin ratios: 3 women and 2 men, 62–79 years; low myosin:actin ratios: 3 women and 1 man, 41–79 years) and 5 control immobilized ICU patients with lesions in the central or peripheral nervous system (2 women and 3 men, 56–70 years). All ICU patients with CIM had been mechanically ventilated for longer than two weeks at the time of muscle biopsy. The biopsies are from patients recruited in a study aiming at improving the diagnostic precision of CIM, i.e., comparing results from percutaneous muscle biopsies with results from biopsies obtained with a disposable microbiopsy instrument. Thus, ICU patients were included who had developed a general paralysis in response to long-term mechanical ventilation and were referred to the clinic for diagnostic purposes. Control samples displaying high myosin:actin ratios were obtained on the first day on the ventilator prior to any myosin loss. In these patients, a myosin loss was evident first after more than one week of immobilization and mechanical ventilation. There is no indication that the inclusion of these patients may have introduced any bias which may have impacted on the results. Each biopsy was frozen in liquid propane chilled by liquid nitrogen and stored at −140 °C or −160 °C until analyses. Written informed consent was obtained from patients' close relatives prior to the study. The study was approved by the Ethical Committee on Human Research at Karolinska Institutet, Stockholm and Uppsala University Hospital, Sweden (approval number, 2016/242-31/2). Informed consent was obtained from all patients or close relatives (deeply sedated and mechanically ventilated ICU patients).

**Statistical analysis**. Data organization, scientific graphing and statistical analyses were performed using Microsoft Excel (2013) and GraphPad Prism (version 7). Results are presented as the mean ± SD, mean ± SEM, or as the difference between means ± SEM, as indicated. Normality was assessed using the Shapiro–Wilk test. Significant differences between two groups were determined by two-tailed Student's *t*-test or Mann–Whitney test and differences with p-values lower than 0.05 were considered significant. Variance analyses of repeated measurements were performed by repeated measures two-way ANOVA, followed by Sidak's test for multiple comparisons; here, differences with *p* values lower than 0.01 were considered significant. Sample size was estimated based on sample availability, previous experimental studies performed in our laboratory, and the literature. Experiments were not randomized or blinded.

**Reporting summary**. Further information on research design is available in the Nature Research Reporting Summary linked to this article.

## Data availability

All data pertaining to this work are shown in the text, figures, and Supplementary Information. The source data underlying Figs. 1e, 2a, b, d, e, 3c, g, 4b–d, 5a–d, 6b, d, 7b–d, 8a, d, and supplementary figs. 2e, 5b, 6b–d, and 7 are provided as a Source Data file. Full scans of images shown in panels 1c, d, 5b, d, 6a, c, 7a, c, d, 8b–d, and supplementary figs. 2b, d, 6c, 8a, b, and 9e are also provided as a separate Source Data file. Source data are provided with this paper.

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

## Acknowledgements

We thank Susanne Kreutzer (Bad Nauheim) for blastocyst injection, Sonja Krüger (Bad Nauheim) for ES cell propagation, Michael Yekelchyk (Bad Nauheim) for sharing expression data of muscle satellite cells, Franziska Koser (Münster) for help with titin gels, and Lisa Kümper (Bochum), Dieter Fürst (Bonn), Wolfgang Obermann (Bochum) and Ralph Knöll (Stockholm) for sharing antibodies. This work was supported by grants from the German Research Foundation (SFB1002, TPA08) and the IZKF Münster (Li1/029/20) to W.A.L. and the FoRUM Research Funding of the Medical Faculty, Ruhr University Bochum (F928R-2018) to S.S.

## Author contributions

S.S. conceived the study approach and design, carried out most of the experiments, analyzed data, wrote a manuscript draft, and generated figures. A.U. performed EM and immuno-EM. Y.L. performed AFM-based nanoindentation and myofiber force measurements. A.V. and M.v.F.-S. performed PCRs, agarose and protein gel electrophoresis and western blotting. A.S., N.C., and L.L. sampled muscle biopsies from CIM patients. T.B. and L.L. provided advice on study design and made revisions to the manuscript. All authors commented on the manuscript. W.A.L. refined the study concept, analyzed data, finalized the manuscript, generated figures, and provided funding.

## Funding

## Competing interests

The authors declare no competing interests.
