## [Peer Review File · Nature Communications]

Reviewers' Comments:

Reviewer #1:

Remarks to the Author:

This work describes a mouse model in which the Z-disc-anchored TTN is inducibly depleted in skeletal muscles. The authors describe that the inactivation of TTN causes sarcomere disassembly and Z-disc deformations, force impairment, myocyte de-stiffening, upregulation of TTN-binding mechanosensitive proteins and activation of protein quality-control pathways, concomitant with preferential loss of thick-filament proteins. Expression of the Cronos-isoform of TTN, generated from an alternative promoter is not affected by the targeting strategy. The authors show that TTN is required to integrate Z-disc and A-band proteins into the mature sarcomere, a that this function is lost when TTN expression is lowered by induction. The group has thus constructed a useful doxycycline inducible, TTN knockout (KO) mouse model targeting the TTN isoforms expressed from the canonical promoter in adult skeletal muscles. The authors also propose that the loss of Z-disc-anchored TTN recapitulates muscle remodelling in critical illness "myosinopathy" patients, characterized by TTN-depletion and loss of thick filaments.

The work is reasonably well and reliably done and the constructed inducible KO mouse model is an improvement and may be a useful model for titinopathies and other muscular disorders in humans. The authors propose that direct previous evidence is limited concerning the requirement of titin for sarcomeric integrity in mature myocytes. Indications for this has however been shown and discussed before in various reports. There are also previous reported titinopathy mouse models although not doxycycline inducible like the described model here. However doxycyclin inducible mousemodels for other genes have been reported before.

Reviewer #2:

Remarks to the Author:

The manuscript by Swist et. al. uses a conditional titin knockout to investigate titin's role in preserving sarcomeric integrity. The well written manuscript shows successful generation of the animal model with evidence of muscle atrophy. The conditional loss of titin disrupts sarcomeres after a few weeks and lowers fiber stiffness. Titin loss is associated with myosin loss as well as the activation of other degradation pathways. This paper provides clear evidence for a functional role in sarcomere maintenance for titin. This work also provides insight into mechanisms by which titin loss leads to atrophy and highlights intriguing similarities with critical illness myosinopathy. However, questions remain regarding the mechanisms at play and their time-course, the effect on function, and quantification. These questions are detailed below:

All of the molecular data appears to be at end-stage of muscle atrophy and mouse viability. The lack of a time course makes it more difficult to interpret molecular mechanisms at play in atrophy of muscles losing titin. An early time point or even a series would be quite helpful in determining what the initiating factors are.

The hang test is a useful measure of muscle function in that it is easily conducted longitudinally. However, it is susceptible to inputs other than direct muscle function, particularly central drive and behavior. It would be helpful to have a direct test of muscle function, such as maximum isometric tension to determine if function is disrupted as much as would be expected from the disordered structure. This would allow correlation between parameters to gain insight into potential mechanisms. The fiber stiffness is also a functional parameter and provides useful information on the state of the cell. However, titin is thought to bear load in tension along the direction of the fibers, which is tested by AFM lateral compression. A passive length tension curve of fibers with and without titin would be more informative than the AFM study. If isolated muscle testing of active mechanics is conducted, whole muscle passive mechanics, with the role of titin would be highly informative to more closely investigate its debated contribution to whole muscle passive

mechanics.

While the images make it apparent that the sarcomeric structure is disordered quantifying this key result would be helpful in relating to the degree and relation to other molecular and functional parameters to help dissect relationships. This could be as simple as intact sarcomere's per field, or more descriptive measures of Z-line shifts in some cases. There are a number of other parameters that could be quantified as well. Muscle atrophy is highlighted by pictures of the hindlimbs, but muscle weights would be informative to see what percent of body mass atrophy is from muscle loss. Quantification of fiber size is helpful, but including percentage of centrally nucleated fibers would add more context to the condition. Quantifying the percent of fibers as in Fig 3 that have disrupted sarcomeres would add a degree to the completeness of the model.

Centrally nucleated fibers are generally a sign of a regenerating fiber rather than a atrophy, while damage and regeneration are often coupled as in dystrophies. It seems likely that satellite cells are activated to try and maintain muscle in the conditional knockout, but that titin is not eliminated until they become mature. Aside from quantifying centrally nucleated fibers, it would be especially interesting to cross the titin mutation onto the commonly used inducible satellite cell (Lepper C & Fan CM, Genesis. 2010) and investigate muscle regeneration following acute damage such as cardiotoxin lacking full-length titin.

Minor comments:

Sup Fig 2D: Considerations of why there is expression of the mutant allele of titin in the lung (pulmonary vein) would be helpful?

Fig 1F: Why is the Wt+dox in cross-section (or oblique) when the Mut+dox appears to be a longitudinal section? Does this change the visibility of the riboprobe?

Fig 2C: There appears to be excessive space between wildtype fibers. Could this be due to paraffin embedding rather than the more typical cryosectioning of muscle?

Fig 3F: The first mutant seems that it should have a similar scale to the wt muscle so that the similar structure can be appreciated.

Fig 3F It was stated that there was significant heterogeneity within fiber in terms of sarcomeric integrity. Is that due to heterogenous recombination within a multinucleated fiber?

Fig 4: Some mutant fibers maintain relatively high stiffness. Where striation patterns observed in those fibers (indicative of preserved titin)? If so that could support the hypothesis of the stiffness relation even within a muscle.

Fig 5: The variability in Chronos is quite stark as alluded to. Some explanation of why would be helpful. Does it try to compensate early and then dissipate later on? Is there any difference between mice who progressed farther within the study vs those that were sacrificed earlier?

Discussion: Please provide a citation for the titin half-life of 2-3 weeks. Previous work has cited 2-3 days in various systems (Isaacs W.B., et al. J. Cell Biol. 1989 and da Silva Lopes K., et al. J. Cell Biol. 2011).

Discussion: Is there any evidence that titin loss precedes the myosin/thick filament loss in ICU patients?

Discussion: It isn't clear to me why the titin mutation has to be in the Z-disc anchoring portion while leaving Chronos intact if it is inducible and skeletal muscle specific?

Discussion: Titin as a contributor to does not appear to be confirmed as a direct contributor to viscoelasticity in this study, although it is likely. Other structures were disrupted secondary to titin deletion that could be altering viscoelasticity within the muscle.

Point-by-point response to referees' comments:

We thank the two reviewers for their insightful comments.

Reviewer #1 (Remarks to the Author):**General comments:**

*"The work is reasonably well and reliably done and the constructed inducible KO mouse model is an improvement and may be a useful model for titinopathies and other muscular disorders in humans. The authors propose **that direct previous evidence is limited concerning the requirement of titin for sarcomeric integrity in mature myocytes**. Indications for this has however been shown and discussed before in various reports. There are also previous reported titinopathy mouse models although not doxycycline inducible like the described model here. However doxycyclin inducible mouse models for other genes have been reported before."*

We are delighted that our report is seen as a valuable contribution to enhance knowledge about titinopathies. We apologize for being not as clear as we should have been about the novelty of our findings compared to relevant previous publications. In the revision, we provide a brief description (already in the Introduction) of what has been reported elsewhere with regard to TTN-deletion models (page 3, paragraph 2). We are well aware that various other models mimic titinopathies, but would like to stress that our mouse model is the first one that demonstrates the structural and functional consequences of an inducible titin deletion in adult muscles. Importantly, our work on muscles from critically illness myopathy patients is unique with regard to the study of titin protein changes, also in relation to other sarcomere protein alterations. We would also like to point out that our study is the first to measure the expression of Cronos protein in skeletal muscle, resulting in the interesting finding that Cronos is expressed in heart and fast-type, but not fatigue-resistant, mouse skeletal muscles.

Reviewer #2 (Remarks to the Author):**General comments:**

*"The manuscript by Swist et. al. uses a conditional titin knockout to investigate titin's role in preserving sarcomeric integrity. The well written manuscript shows successful generation of the animal model with evidence of muscle atrophy. The conditional loss of titin disrupts sarcomeres after a few weeks and lowers fiber stiffness. Titin loss is associated with myosin loss as well as the activation of other degradation pathways. This paper provides **clear evidence** for a functional role in sarcomere maintenance for titin. This work also provides insight into mechanisms by which titin loss leads to atrophy and highlights intriguing similarities with critical illness myosinopathy. However, questions remain regarding the mechanisms at play and their time-course, the effect on function, and quantification. These questions are detailed below:"*

Thank you for your constructive comments. Following your suggestions, we have added new data on the properties and function of TTN-mutant mice that were exposed to doxycycline for a shorter period of time than before (3 weeks compared to 4-5 weeks). A detailed explanation of the changes in the manuscript is supplied below.

Comment 1: *“All of the molecular data appears to be at end-stage of muscle atrophy and mouse viability. The lack of a time course makes it more difficult to interpret molecular mechanisms at play in atrophy of muscles losing titin. An early time point or even a series would be quite helpful in determining what the initiating factors are.”*

We agree that the lack of a time course of changes was a shortcoming of the initial manuscript version, although the 1-month-long treatment with doxycycline did not yet cause end-stage muscle atrophy and mouse viability in this model. However, we were asked by the local animal care and use committee to stop the experiment when our mutant mice reached 20% weight loss. In this revision, we have added data on mutant mice treated with doxycycline for only 21 days, resulting in 0-10% body weight loss (Fig. 2d, e; Fig. 3c, Suppl. Fig. 5a, b; new Suppl. Fig. 6). Importantly, at this earlier time point, we already observe changes in titin expression and sarcomere structure, albeit not as prominent as in mice exposed to doxycycline for longer times. A completely new figure reporting expression changes of titin and other relevant molecules at the earlier time-point has been added (Suppl. Fig. 6).

Comment 2: *“The hang test is a useful measure of muscle function in that it is easily conducted longitudinally. However, it is susceptible to inputs other than direct muscle function, particularly central drive and behavior. It would be helpful to have a **direct test of muscle function**, such as **maximum isometric tension** to determine if function is disrupted as much as would be expected from the disordered structure. This would allow correlation between parameters to gain insight into potential mechanisms. The **fiber stiffness** is also a functional parameter and provides useful information on the state of the cell. However, titin is thought to bear load in **tension along the direction of the fibers**, which is tested by AFM lateral compression. **A passive length tension curve of fibers with and without titin would be more informative than the AFM study.** If isolated muscle testing of active mechanics is conducted, whole muscle passive mechanics, with the role of titin would be highly informative to more closely investigate its debated contribution to whole muscle passive mechanics.”*

Thank you for this comment. Following the suggestions, we have added data on both passive tension and Ca²⁺-activated tension of wildtype and TTN-depleted muscle fibers (Fig. 4 c, d). The new results show a dramatic loss in passive and active force after inactivation of titin.

Comment 3: *“While the images make it apparent that the sarcomeric structure is disordered quantifying this key result would be helpful in relating to the degree and relation to other molecular and functional parameters to help dissect relationships. This could be as simple as intact sarcomere’s per field, or more descriptive measures of Z-line shifts in some cases. There are a number of other parameters that could be quantified as well. Muscle atrophy is highlighted by pictures of the*

hindlimbs, but muscle weights would be informative to see what percent of body mass atrophy is from muscle loss.”

We agree with this suggestion and thus counted myocytes with intact or disrupted sarcomeres (per field-of-view) in mice that had lost either <10% or 10-20% of their initial body weight due to the TTN deficiency (Fig. 3c). Additionally, we now report muscle weight loss, in addition to body weight loss, in wildtype and mutant mice after 21 days of doxycycline treatment (Suppl. Fig. 5b). These new data demonstrate that muscle is wasted before reductions in body weight become obvious.

Comment 4: *“Quantification of fiber size is helpful, but including percentage of centrally nucleated fibers would add more context to the condition. Quantifying the percent of fibers as in Fig 3 that have disrupted sarcomeres would add a degree to the completeness of the model. Centrally nucleated fibers are generally a sign of a regenerating fiber rather than a atrophy, while damage and regeneration are often coupled as in dystrophies.”*

Thank you for this comment. Following the suggestion, we have counted myofibers with centralized nuclei in doxycycline-treated WT and MUT (0-10% and 10-20% body weight loss, respectively) mouse muscles and included the data in the revised manuscript (Fig. 2d). A maximum of only 1.5% of cells presented with centralized nuclei in the MUT mice, indicating limited muscle regeneration. We have also added quantification of myofiber cross-sectional area for the 21-d-doxycycline mice (Fig. 2e).

Comment 5: *“It seems likely that satellite cells are activated to try and maintain muscle in the conditional knockout, but that titin is not eliminated until they become mature. Aside from quantifying centrally nucleated fibers, it would be especially interesting to cross the titin mutation onto the commonly used inducible satellite cell (Lepper C & Fan CM, Genesis. 2010) and investigate muscle regeneration following acute damage such as cardiotoxin lacking full-length titin.”*

Based on the low number of myofibers with centralized nuclei we can exclude that regeneration is a major contributor to the observed phenotype. According to Li et al, *BMC Dev Biol.* 2015;15:42 (PMID: 26559169), the 5-day-doxycycline treatment of an Acta1-TetO_Cre reporter mouse leads to robust transgene expression and only a very modest mosaicism. In our experiments, we kept the mice under constant doxycycline treatment, making it very unlikely that myofibers with regular sarcomeres re-express TTN. Moreover, the constitutive TTN-KO demonstrated the absence of organized sarcomeres in the developing embryo, underscoring that Z-disc-anchored titin is necessary for sarcomere assembly. We attribute the relatively late onset of sarcomere disassembly in our mutants (after 21d of dox-treatment, ~80% of fibers had yet-undisturbed sarcomere structure; Fig. 3c) to the long half-life of the giant titin protein, which was reported to be 2-3 weeks in adult mice (Fornasiero et al. *Nat Commun.* 2018;9:4230. PMID: 30315172). This paper is now cited in the manuscript. Please see also our answer to minor comment H).

While it may potentially be interesting to cross our mouse model with Pax7-CreERT2 mice (Lepper C & Fan CM, *Genesis.* 2010) and investigate muscle regeneration following acute damage such as cardiotoxin, we consider such an experiment to be out of the scope of our current paper. In this context, the group of co-author Thomas Braun has analyzed the expression profiles of quiescent, proliferating and differentiated satellite cells by RNA-sequencing (unpublished data) and found no

expression of titin, novex-3, myosin heavy chain 3 (Mhy3) and skeletal muscle actin alpha 1 (Acta1) in freshly isolated satellite cells. In proliferating satellite cells, the expression of these genes was increased to a minor extent but only reached high levels during differentiation. Examples for these findings are included below. Since Ttn and Acta1 are concomitantly up-regulated with satellite cell differentiation, we can conclude that mutant Ttn first occurs early during differentiation in our model but does not affect quiescent satellite cells. It is certainly interesting to study the relevance of TTN in proliferating satellite cells for muscle homeostasis in normal and diseased states. However, answering this exciting question was not the aim of this work.

Minor comments:

A: "Sup Fig 2D: Considerations of why there is expression of the mutant allele of titin in the lung (pulmonary vein) would be helpful?"

Cardiomyocytes are present in the lung, specifically along the pulmonary veins, as we have also pointed out in the manuscript (Supplemental file, p. 7) (please also see the lacZ staining in Suppl. Fig. 3). In Suppl. Fig. 2d, the RT-PCR amplifies exons 1 to 8 (1.4 kb) of the full-length *Ttn* message expressed from the WT allele of a heterozygous embryo and wildtype adult tissue. However, there is no full-length *Ttn* message detectable in homozygous mutant *Ttn*^{tm1a} embryos.

B: "Fig 1F: Why is the Wt+dox in cross-section (or oblique) when the Mut+dox appears to be a longitudinal section? Does this change the visibility of the riboprobe?"

Thank you for this hint. We have exchanged this image to clarify the situation. However, since in our case the obliqueness of the section was not severe, we did not observe that the sectioning plane significantly influenced visibility of the riboprobe.

C: "Fig 2C: There appears to be excessive space between wildtype fibers. Could this be due to paraffin embedding rather than the more typical cryosectioning of muscle?"

The images shown in Fig. 2c are from cryosections. For cryo-embedding, we fixed muscles in 4% paraformaldehyde at 4°C. Subsequently, tissues were incubated first in 15% sucrose/PBS, then in

30% sucrose in PBS at 4°C, until the tissue had sunk down. Next, tissues were embedded in OCT and cooled on dry ice. We extended the “Methods” section to describe this procedure.

With regard to the space between muscle fibers, incomplete fixation potentially might indeed lead to cell shrinkage during the procedure. We processed the MUT and WT tissues from gender and age-matched littermates always concurrently, to avoid differences due to processing of material.

D: *“Fig 3F: The first mutant seems that it should have a similar scale to the wt muscle so that the similar structure can be appreciated.”*

The image and scale in this figure (new Fig. 3G) were adjusted accordingly.

E: *“Fig 3F It was stated that there was significant heterogeneity within fiber in terms of sarcomeric integrity. Is that due to heterogenous recombination within a multinucleated fiber?”*

Differences among nuclei in single muscle cells have been detected before, which likely account for our observations. These nuclei differ in transcriptional activity and protein accumulation (Cutler et al., J Cell Sci. 2018;131(3); PMID: 29361530). Furthermore, it is assumed that within one multinucleated myotube, individual nuclei determine the protein synthesis for a finite volume of cytoplasm (Schwartz, Front Physiol. 2019; 9, 1887. PMID: 30740060). Thus, significant heterogeneity within a fiber is expected in our MUT model.

F: *“Fig 4: Some mutant fibers maintain relatively high stiffness. Where striation patterns observed in those fibers (indicative of preserved titin)? If so that could support the hypothesis of the stiffness relation even within a muscle.”*

Considering the heterogeneity in structural remodeling among different MUT myofibers, substantial heterogeneity can also be expected for functional parameters, which was indeed what we observed. Moreover, the data scatter observed in the AFM nanoindentation measurements is typical for this type of experiment, even in wildtype cells. For instance, the apparent stiffness depends on the local cellular substructure below the point of indentation.

Striation patterns were occasionally observed in MUT fibers.

G: *“Fig 5: The variability in Cronos is quite stark as alluded to. Some explanation of why would be helpful. Does it try to compensate early and then dissipate later on? Is there any difference between mice who progressed farther within the study vs those that were sacrificed earlier?”*

We were also intrigued by this observation. However, we could not find a statistically significant correlation between Cronos expression and body weight loss/dox-treatment duration. The upregulation of Cronos seen in some MUT muscles was the exception rather than the rule. The newly added data from MUT mice after 21d of dox-treatment again demonstrate unaltered Cronos on protein and RNA level (Suppl. Fig. 6).

H: *“Discussion: Please provide a citation for the titin half-life of 2-3 weeks. Previous work has cited 2-3 days in various systems (Isaacs W.B., et al. J. Cell Biol. 1989 and da Silva Lopes K., et al. J. Cell Biol. 2011).”*

We apologize for the missing reference, which is now added to the revised manuscript. The cited work by Fornasiero et al. *Nat Commun.* 2018;9:4230 (PMID: 30315172) used *in vivo* isotopic labeling and mass spectrometry to measure the accurate lifetimes of murine brain proteins but also titin in

cardiac and gastrocnemius muscles. In their Supplementary data section, they reported a titin half-life of 9.18 days in adult heart and 23.92 days in adult gastrocnemius muscle.

The shorter titin-life reported in the article by *Isaacs et al. J. Cell Biol. 1989* refers to the protein half-life in synchronized cultures of skeletal muscle cells derived from day 12 chicken embryos. The half-life of titin presumably slows down significantly with age.

The experimental approach of *da Silva Lopes et al. J. Cell Biol. 2011* included a titin-eGFP knockin mouse to analyze titin motility by measuring fluorescence recovery after photobleaching in embryonic and neonatal cardiomyocytes. However, the recovery rates after photobleaching does not indicate the lifetime of titin. Recovery rates rather provide information about the exchange of titin between the cytosolic and sarcomeric protein pools. Please also see a recent report (*Cadar et al. Am J Physiol Cell Physiol. 2020 318(1):C163-C173*) about extensive reversible photobleaching, the presence of which obscures interpretation of such experiments.

I: *“Discussion: Is there any evidence that titin loss precedes the myosin/thick filament loss in ICU patients?”*

We propose such a scenario at the end of our discussion, based on the similarities found between our titin MUT model and the CIM patient muscles. However, there is no definitive evidence for this assumption, which we now indicated in the discussion.

J: *“Discussion: It isn’t clear to me why the titin mutation has to be in the Z-disc anchoring portion while leaving Chronos intact if it is inducible and skeletal muscle specific?”*

To avoid concomitant targeting of Cronos and full-length Titin, the mutation has to be placed upstream of the alternative Cronos promoter in titin intron 239. However, the exons upstream of the Cronos promoter encode for the titin I-band region, which is known to be extensively spliced. Thus, insertion of the mutation into I-band TTN bears the risk of generating a hypomorphic phenotype. Therefore, the Z-disc portion of titin was the logical target for the mutation to test our hypothesis that full-length titin is crucial for adult sarcomere integrity.

K: *“Discussion: Titin as a contributor to does not appear to be confirmed as a direct contributor to viscoelasticity in this study, although it is likely. Other structures were disrupted secondary to titin deletion that could be altering viscoelasticity within the muscle.”*

The main goal of our mouse model was to address the role of titin stiffness for sarcomere maintenance but not for overall muscle elasticity and passive stiffness. The mechanical data obtained on WT and MUT fibers do support an important function of titin in passive and active tension generation. However, other approaches are needed to more directly test this issue, such as the acute clipping of the titin spring, as we have done recently (<https://www.biorxiv.org/content/10.1101/577445v2>).

Response to Editors comments:

“In particular, we would ask a revised manuscript to address the concerns of reviewer #1 regarding the novelty of your study by including an in-depth and balanced discussion of previous related

findings, highlighting the advance provided by your own work in the context of these other studies. In addition, we would ask a revised manuscript to address all the concerns raised by reviewer #2.”

We thank the Editors for handling this manuscript and allowing us to address the comments of the reviewers in a revised manuscript. We have responded to the concerns of reviewer #1 regarding the novelty of our study by stating (already in the Introduction) the uniqueness of our approach versus previous papers reporting titin deletion or knockout models. Our paper is the first to report TTN inactivation in adult muscles. Moreover, we show for the first time a striking loss of titin in critical illness myopathy, along with the known loss of thick filament proteins in this “myosinopathy”.

We have also responded in full to the comments of reviewer #2. In particular, we have studied our mutant mice at an earlier time-point of TTN inactivation, after 21 days of dox-treatment. We provide new data for this stage in the following figures: Fig. 2d, e; Fig. 3c, g, Fig. 4c, d, Suppl. Fig. 5a, b, and Suppl. Fig. 6 (new). We find that titin is reduced already after 21 days of dox-treatment causing some sarcomere disruption, but these effects involve much less cells than after 30 days of dox-treatment. Relevant text changes can be appreciated in the “highlighted changes” manuscript file. We have also discussed this reviewer’s suggestion to cross the titin mutation onto an inducible satellite cell model and investigate muscle regeneration following acute muscle damage. Although we believe that this request falls outside of the scope of this paper, we have provided unpublished data on sarcomere transcript expression in quiescent and activated satellite cells in our response, which we hope the reviewer might find interesting in the context of this discussion. All other reviewer requests have also been answered, as can be seen in our point-by-point response.

Reviewers' Comments:

Reviewer #1:

Remarks to the Author:

The authors propose that direct previous evidence is limited concerning the requirement of titin for sarcomeric integrity in mature myocytes. Indications for this has however been shown and discussed before in various reports. There are also previous reported titinopathy mouse models although not doxycycline inducible like the described model here. Doxycyclin inducible mousemodels for other genes have been reported before and could be cited.

This seems to be an useful model for studying titin depletion, and the revised manuscript has improved.

Suggested changes:

Abstract rows 28-29:

The giant protein titin is claimed to be required for sarcomeric integrity in mature myocytes, but direct evidence for this hypothesis is limited.

Could be changed to:

"The giant protein titin is reported to be required for sarcomeric integrity in mature myocytes, but reported details of this is limited."

Introduction

First part rows 50-52

The following reference could be cited and from this some small addition from it to the text:
The complexity of titin splicing pattern in human adult skeletal muscles. Savarese M et al Skelet Muscle. 2018 Mar 29;8(1):11

Row 59-62

It has also been speculated by many that TTN is relevant for the maintenance of the mature sarcomere. However, this function has been more difficult to assess and has remained uncertain, because TTN is necessary for cardiac development and its absence leads to early embryonic lethality^{12,13,17}.

Could be changed to:

It has been suggested in many reports that TTN is relevant for the maintenance of the mature sarcomere. This function has been more difficult to assess and details remained unclear. However TTN is necessary for cardiac development and its absence leads to early embryonic lethality^{12,13,17}.

In the part rows 63-76 the below mouse model could be mentioned carrying the FINmaj mutation in TTN. The mutation leads to the loss of the very C-terminal end of titin and to a secondary deficiency of calpain 3, a partner of titin.

Removal of the calpain 3 protease reverses the myopathology in a mouse model for titinopathies. Charton K, et al. Hum Mol Genet. 2010 Dec 1;19(23):4608-24.

Discussion part:

Rows 310-111

Surprisingly little is known about the life cycle of adult muscle sarcomeres, including their maintenance and turnover of proteins.

Could be changed to:

Not all details are known about the life cycle of adult muscle sarcomeres, including their maintenance and turnover of proteins.

Reviewer #2:

Remarks to the Author:

The revised manuscript by Swist et. al. enhances the understanding of titin's role in sarcomere maintenance with a variety of experiments. The addition of an earlier time point is helpful as is the additional mechanical data in understanding the disruption of previously intact sarcomeres when titin is lost. Further quantification of parameters of disruption clearly demonstrate the progression. The evidence of early expression of titin along with acta1 implies that titin would be eliminated early in myoblast differentiation and while could be interesting, is outside the scope of the current manuscript. Previous concerns have been well addressed by the revision. The manuscript provides novel data on titin's role in sarcomere maintenance (as opposed to sarcomerogenesis with previous models) with a new model that can be further employed to examine the roles of titin with greater control.

Point-by-point response to referees' comments:**Reviewer #1 (Remarks to the Author):**

"The authors propose that direct previous evidence is limited concerning the requirement of titin for sarcomeric integrity in mature myocytes. Indications for this has however been shown and discussed before in various reports. There are also previous reported titinopathy mouse models although not doxycycline inducible like the described model here. Doxycyclin inducible mouse models for other genes have been reported before and could be cited.

This seems to be an useful model for studying titin depletion, and the revised manuscript has improved."

We appreciate the comments of this reviewer, thank you.

"Suggested changes:

Abstract rows 28-29:

The giant protein titin is claimed to be required for sarcomeric integrity in mature myocytes, but direct evidence for this hypothesis is limited.

Could be changed to:

"The giant protein titin is reported to be required for sarcomeric integrity in mature myocytes, but reported details of this is limited."

Response: We have edited the first abstract sentence accordingly.

"Introduction

First part rows 50-52

The following reference could be cited and from this some small addition from it to the text:

The complexity of titin splicing pattern in human adult skeletal muscles. Savarese M et al Skelet Muscle. 2018 Mar 29;8(1):11"

Response: We have added the citation suggested.

"Row 59-62

It has also been speculated by many that TTN is relevant for the maintenance of the mature sarcomere. However, this function has been more difficult to assess and has remained uncertain, because TTN is necessary for cardiac development and its absence leads to early embryonic lethality^{12,13,17}.

Could be changed to:

It has been suggested in many reports that TTN is relevant for the maintenance of the mature sarcomere. This function has been more difficult to assess and details remained unclear. However TTN is necessary for cardiac development and its absence leads to early embryonic lethality^{12,13,17}."

Response: We have edited this text accordingly.

“In the part rows 63-76 the below mouse model could be mentioned carrying the FINmaj mutation in TTN. The mutation leads to the loss of the very C-terminal end of titin and to a secondary deficiency of calpain 3, a partner of titin.

Removal of the calpain 3 protease reverses the myopathology in a mouse model for titinopathies. Charton K, et al. Hum Mol Genet. 2010 Dec 1;19(23):4608-24.”

Response: We have added the citation suggested.

“Discussion part:

Rows 310-111

Surprisingly little is known about the life cycle of adult muscle sarcomeres, including their maintenance and turnover of proteins.

Could be changed to:

Not all details are known about the life cycle of adult muscle sarcomeres, including their maintenance and turnover of proteins.”

Response: We have edited this text accordingly.

Reviewer #2 (Remarks to the Author):

“The revised manuscript by Swist et. al. enhances the understanding of titin’s role in sarcomere maintenance with a variety of experiments. The addition of an earlier time point is helpful as is the additional mechanical data in understanding the disruption of previously intact sarcomeres when titin is lost. Further quantification of parameters of disruption clearly demonstrate the progression. The evidence of early expression of titin along with acta1 implies that titin would be eliminated early in myoblast differentiation and while could be interesting, is outside the scope of the current manuscript. Previous concerns have been well addressed by the revision. The manuscript provides novel data on titin’s role in sarcomere maintenance (as opposed to sarcomerogenesis with previous models) with a new model that can be further employed to examine the roles of titin with greater control.”

We appreciate the comments of this reviewer, thank you.